# TGFβ promotes widespread enhancer chromatin opening and operates on genomic regulatory domains

Jose A. Guerrero-Martínez[1], María Ceballos-Chávez [1], Florian Koehler[2,3], Sandra Peiró[4] & Jose C. Reyes [1]✉

The Transforming Growth Factor-β (TGFβ) signaling pathway controls transcription by regulating enhancer activity. How TGFβ-regulated enhancers are selected and what chromatin changes are associated with TGFβ-dependent enhancers regulation are still unclear. Here we report that TGFβ treatment triggers fast and widespread increase in chromatin accessibility in about 80% of the enhancers of normal mouse mammary epithelial-gland cells, irrespective of whether they are activated, repressed or not regulated by TGFβ. This enhancer opening depends on both the canonical and non-canonical TGFβ pathways. Most TGFβ-regulated genes are located around enhancers regulated in the same way, often creating domains of several co-regulated genes that we term TGFβ regulatory domains (TRD). CRISPR-mediated inactivation of enhancers within TRDs impairs TGFβ-dependent regulation of all co-regulated genes, demonstrating that enhancer targeting is more promiscuous than previously anticipated. The area of TRD influence is restricted by topologically associating domains (TADs) borders, causing a bias towards co-regulation within TADs.

[1] Centro Andaluz de Biología Molecular y Medicina Regenerativa-CABIMER, Consejo Superior de Investigaciones Científicas-Universidad de Sevilla-Universidad Pablo de Olavide (CSIC-USE-UPO), Avenida Americo Vespucio 24, 41092 Seville, Spain. [2] Division of Epigenetics, DKFZ-ZMBH Alliance, German Cancer Research Center, Heidelberg, Germany. [3] Faculty of Biosciences, Heidelberg University, Heidelberg, Germany. [4] Vall d'Hebron Institute of Oncology (VHIO), 08035 Barcelona, Spain. ✉email: jose.reyes@cabimer.es

TGFβ and related ligands are believed to be the family of cytokines with the most diverse and widespread functions in metazoans[1]. TGFβ cytokines have crucial roles in development, proliferation, tissue homeostasis, differentiation, and immune regulation. Consequently, alterations in TGFβ signaling underlie numerous diseases, including cancer[1,2]. The TGFβ canonical pathway involves ligand binding to a transmembrane tetrameric complex of dual-specificity kinase receptors, which becomes activated and can phosphorylate receptor-regulated (R-) SMAD proteins (SMAD2 and SMAD3, among others) at carboxy-terminal serine residues[3]. R-SMAD factors shuttle between the nucleus and the cytoplasm; upon phosphorylation, they oligomerize together with SMAD4 and accumulate in the nucleus, where they bind chromatin[4]. Chromatin immunoprecipitation (ChIP)-seq experiments demonstrated that SMAD proteins mostly bind to distant cis-regulatory regions, commonly called enhancers[5–9]. Enhancers are regions for sequence-specific transcription factors (TFs) binding, which potentiate transcription of their target genes, independently of their relative distance, location or orientation respective to them[10–13]. The human genome has more than one million enhancers; however, only a subset of these are accessible and, eventually, active in each cell-type. Chromatin can regulate enhancer function by allowing or preventing TF binding to its target motifs within an enhancer. However, the so-called pioneer TFs can bind DNA that is wrapped around a histone octamer and trigger enhancer regulation by recruiting chromatin remodeling complexes and histone modifying enzymes[14]. The current model proposes that most lineage-determining TFs (LDTFs) are pioneer TFs that cooperate to establish the cell-type-specific landscape of open enhancers[5,6,10,15]. This mechanism would restrict the number of enhancers at which signal-driven TFs (SDTFs) can bind. Nonetheless, de novo enhancer chromatin opening by SDTFs alone, or in combination with LDTFs, have also been reported[16,17]. Thus, how SDTFs lead to enhancer selection and activation is still an open question.

The epithelial-to-mesenchymal transition (EMT) switches cell phenotypes from epithelial to mesenchymal during development[18–20] and has been linked to dissemination and migration of epithelial tumor cells[21] and metastasis formation[22–25]. TGFβ is one of the most potent inductors of EMT in normal and oncogenic epithelial cells from different origins[26]; however, little is known about the genomic repertoire of enhancers activated by TGFβ, the chromatin dynamics during this process or the SMAD partners in epithelial cells.

We have now addressed these outstanding questions about the genomic regulation mediated by TGFβ in epithelial cells by determining and characterizing the enhancer atlas of the TGFβ response in normal murine mammary gland (NMuMG) epithelial cells, a well-established model for TGFβ-dependent EMT[27]. We show that TGFβ promotes a fast and widespread increase of chromatin accessibility (assayed by ATAC-seq, ATAC-see and DNaseI-seq) in most enhancers of the cell line, irrespectively of whether the enhancer will become activated or repressed. Pervasive chromatin opening depends on both the canonical and the non-canonical TGFβ pathways. Activated enhancers are strongly enriched for SMAD2/3/4 and AP-1 footprints. Strikingly, analyses of the regulated genes around TGFβ-regulated enhancers revealed TGFβ regulatory domains that can encompass several genes and that are constrained by three-dimensional (3D) chromatin conformation. In these domains enhancer targeting is more promiscuous than previously anticipated.

## Results

### TGFβ provokes widespread enhancer chromatin opening.
To better understand the mechanism by which TGFβ signaling controls transcription in epithelial cells, we investigated early-chromatin changes provoked by TGFβ treatment. TGFβ signals through several transduction pathways, most of which are stimulated by growth factors present in the serum. Therefore, to clearly determine all the changes promoted by TGFβ, NMuMG cells were depleted of serum and insulin (an additive of the NMuMG cell culture medium) for 6 h, prior to TGFβ1 treatment. Then, samples were taken after 2 h to monitor fast chromatin modifications, and after 12 h (when mesenchymal-like morphological changes are first observed) to monitor late chromatin modifications (Fig. 1a). To systematically identify functional enhancers, we first determined chromatin accessibility by assay for transposase-accessible chromatin followed by high-throughput sequencing (ATAC-seq)[28]. A total of 52,626, 100,459, and 105,334 significant peaks (IDR (irreproducible discovery rate) <0.05) were identified from vehicle, 2 h and 12 h TGFβ samples, respectively (see correlations between replicates, Supplementary Fig. 1a). Peaks with summits closer than 1000 bp were considered as overlapping and merged. All ATAC-seq peaks found in any of the three tested conditions were then merged, resulting in 39,432 regions of open chromatin. Of these, 10,362 were coincident with previously defined transcription start sites (TSS), and 29,071 were considered to be putative enhancers. Notably, TGFβ treatment (at 2 h and 12 h) provokes a widespread increase of chromatin accessibility at both TSS and putative enhancers, concomitant with a decrease of ATAC signal in the rest of the genome (Supplementary Fig. 1b). Pervasive increase of chromatin accessibility was confirmed by DNaseI-seq (Supplementary Fig. 1c, d). Thus, 88.3% of the DNaseI-seq peaks (20,908) were coincident with ATAC peaks and 76.8% of the DNaseI-ATAC common peaks increased their signal at 2 h of TGFβ treatment. Furthermore, a very good correlation was observed between DNaseI-seq and ATAC-seq fold changes (FCs).

We next imaged the transposase-accessible chromatin in situ using ATAC-see[29], which uses microscopy to visualize accessible chromatin at the single-cell level, by inserting fluorophores by the Tn5 transposase at open-chromatin sites. First, we confirmed that TGFβ treatment promoted a general increase of the ATAC-see signal (Fig. 1b). Signal was visible as a general nuclear background, but also concentrated into puncta. TGFβ treatment increased the total intensity of the ATAC-see signal per nuclei by about fourfold, with slightly fewer puncta per nucleus, as shown by 3D confocal microscopy image analysis using TANGO software[30] (Fig. 1c–f). Notably, 2 h of TGFβ treatment strongly increased the volume and fluorescence intensity of the puncta (Fig. 1g, h). Kinetic analysis revealed that general chromatin opening occurred as early as 10 min after TGFβ addition (Fig. 1i, j), coinciding with the reported accumulation of SMAD2/3 and SMAD4 in the nucleus of TGFβ treated cells[31] and with the activation of the ERK pathway (Supplementary Fig. 2a). We verified that TGFβ-dependent widespread chromatin opening also occurred in cells not deprived of growth factors prior to TGFβ stimulation (Supplementary Fig. 2b–e) or with severe serum starvation for 24 h or 48 h prior to TGFβ treatment (Supplementary Fig. 2f). TGFβ-dependent generalized chromatin opening also occurred in other epithelial cell lines, such as the human MCF7 and RPE1 (Supplementary Fig. 2g), although with much slower kinetics, which is consistent with the slower EMT process in these cell lines[32,33]. Critically, knockdown of SMAD4 by either of two different siRNAs abolished the increase of chromatin accessibility (Fig. 1k and Supplementary Fig. 2h). Furthermore, increased accessibility after TGFβ treatment was strongly impaired not only by the ALK4/5/7 TGFβ receptors inhibitor SB431542, but also by the TGFβ−activated kinase 1 (TAK1) inhibitor Takinib[34] or the MEK inhibitor U0126, which prevents ERK activation (Fig. 1l). Thus, both canonical and non-

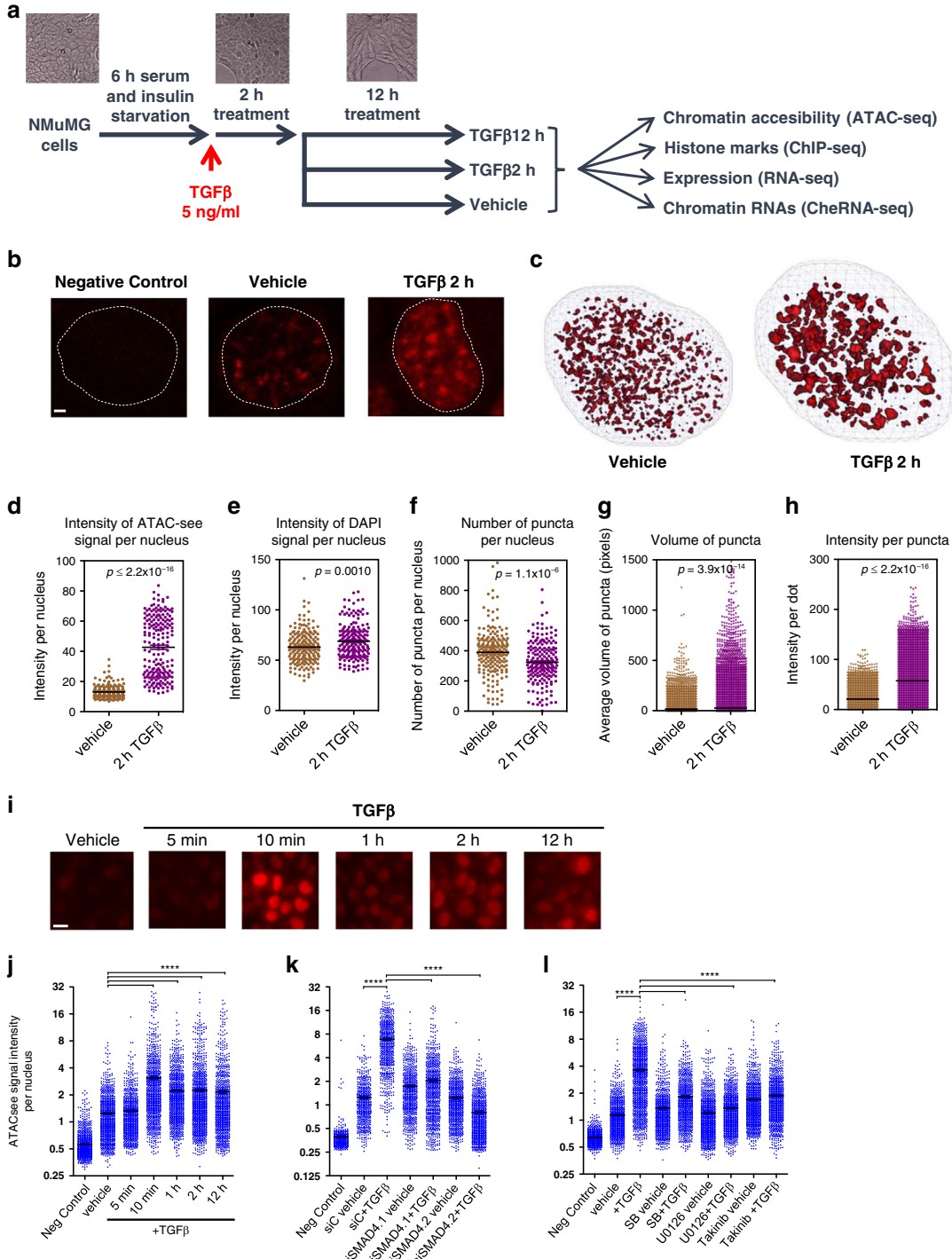

canonical TGFβ pathways are required for widespread chromatin opening.

The putative enhancers were first classified into four categories depending on differential accessibility measured by changes in the average ATAC-seq signal provoked by TGFβ after 2 h (Fig. 2a). TGFβ treatment led to increased accessibility in 75.0% and 79.7% of enhancers at 2 h and 12 h, respectively (linear FC ≥ 1.5-fold). Proportional DNaseI accessibility gain was also observed in the same four categories (Supplementary Fig. 1e). Widespread increases of chromatin accessibility was even more evident at the early timepoint of 10 min, as analyzed by ATAC-seq (88.67% of the peaks had FC ≥ 1.5) (Fig. 2b), confirming the ATAC-see kinetics.

We next investigated the role of TFs in chromatin opening by genomic TF footprint analysis using ATAC-seq data[28], which detects actual TF occupancy to a single-nucleotide resolution, based on the steric hindrance between TF complexes and the Tn5 transposase used for ATAC experiments. AP-1 footprints were strongly overrepresented in most enhancers (Fig. 2c, p-values in Supplementary Fig. 3a). AP-1 is a TF comprising dimers of proteins belonging to the JUN, FOS, ATF, MAF and CREB subfamilies, with important roles in differentiation, apoptosis, and proliferation[35]. Notably, AP-1, SMAD4 and SMAD3 footprints were differentially enriched in enhancers that displayed a strong gain of accessibility (FC ≥ 4) 2 h and 12 h after TGFβ, but not at the 10 min timepoint, which instead were enriched for

**Fig. 1 TGFβ provokes widespread enhancer chromatin opening. a** Flow diagram depicting experimental design. Vehicle-treated cells (control) were taken after 2 h of treatment. **b** Confocal microscope images of ATAC-see signals from NMuMG nuclei of cells cultured under the indicated conditions. Negative control shows ATAC-see signal obtained in the presence of 50 mM EDTA, which inhibits Tn5 transposome activity. Bar, 1 μm. Representative images out of twelve independent experiments are shown. **c** Representative images showing processing of ATAC-see 3D signal by TANGO software in cells treated with TGFβ or vehicle for 2 h. **d–h** Quantification of ATAC-see images by using TANGO software. Scatter dot plots showing ATAC-see signal intensity (**d**), DAPI signal intensity (**e**), and number of puncta per nucleus (**f**) from 203 (vehicle) and 194 (2 h TGFβ) nuclei from two independent experiments. **g, h** Scatter dot plots showing volume of puncta expressed as pixels (**g**), and intensity of ATAC-see signal per puncta (**h**). Number of puncta analyzed: 80,839 for vehicle and 63,001 for 2 h TGFβ from 203 (vehicle) or 194 (2 h TGFβ) cells, from two independent experiments. Two-tailed Mann–Whitney test *p*-values are provided. **i** Fluorescence microscope images of ATAC-see signals from cells cultured under the indicated conditions. Bar, 10 μm. Representative images out of three independent experiments are shown. **j–l** Quantification of ATAC-see signal intensity of nuclei at the indicated times (**j**) or 10 min after TGFβ or vehicle addition (**k, l**). **k** Cells were transfected 48 h before treatment with siControl (siC) or with two different siRNAs against SMAD4 (siSMAD4.1 or siSMAD4.2). **l** Cells were treated with 10 μM of SB431542, U0126, or Takinib for 2 h before TGFβ addition. Merged data from three independent experiments are shown. Statistical significance between indicated samples was determined by using the two-tailed Mann–Whitney test, ****$p \leq 0.0001$. Quantified nuclei (*n*) in each scatter dot plot and exact *p*-values are provided in Supplementary Data 5. **d–h, j–l** The horizontal black line of the scatter dot plots represents the mean value. Source data are provided as a Source Data file.

footprints related to ETS transcription factors (e.g., ELK4, ETV, ELF1, and GABPA) (Fig. 2c–f and Supplementary Fig. 3a, b). Indeed, at the 2 h and 12 h timepoints of TGFβ, SMAD footprints were negatively enriched (underrepresented) in enhancers with little or no change of accessibility (FC < 1.5). These results were confirmed by analyzing recently published ChIP-seq data of SMAD2/3 in NMuMG cells, at 1.5 h after TGFβ-treatment[36] (Supplementary Fig. 3c, d). Many enhancers that strongly gained accessibility at 2 h and 12 h after TGFβ were co-occupied by SMAD2/3 and AP-1, and these displayed about 25% more accessibility than enhancers containing only one of the two factors (Supplementary Fig. 3e, f). Enhancers with little or no change of accessibility (FC < 1.5) at 2 h and 12 h after TGFβ were notably enriched in CTCF and CTCFL1 sites (Fig. 2c, d, f and Supplementary Fig. 3a, b), suggesting that at least part of these accessible regions have a structural function.

Both Jun and Fos, two classical AP-1 factors, were induced by TGFβ after 2 h, but not after 10 min of treatment, as shown by western blotting (Fig. 2g). Furthermore, RNA-seq transcriptomic analysis confirmed that several genes encoding components of AP-1 dimers, including *Jun, Fos, JunB, JunD, Fosl1, Fosl2, Jdp2* and *Atf3*, were also significantly upregulated by 2 h and 12 h of TGFβ treatment (Fig. 2h and Supplementary Fig. 4a, Supplementary Data 1). Thus, TGFβ causes a fast and widespread chromatin opening, which depends on both the canonical and the non-canonical TGFβ pathways and which correlates to increased occupancy of AP-1 and SMAD TFs at enhancers 2 h and 12 h, but not 10 min, after TGFβ addition. In the next sections we will concentrate in the chromatin changes that occurs 2 h and 12 h after TGFβ.

**Identification and characterization of TGFβ-regulated enhancers.** Enhancers are often classified based on their histone post-transcriptional modifications[12,15]. Active enhancers contain histone H3 acetylated at lysine K27 (H3K27ac), histone H3 monomethylated at lysine 4 (H3K4me1)[37–39], and RNA polymerase II which often synthesizes enhancer RNAs (eRNAs)[40]. Primed enhancers contain only H3K4me1 and present a stable pre-activated situation. Finally, latent enhancers are genomic regions that lack enhancer-associated marks (i.e., H3K4me1 and H3K27ac) but that acquire these features upon induction by cellular signaling pathways[17]. We determined H3K4me1 and H3K27Ac modifications genome wide by ChIP-seq in NMuMG cells, under the same conditions and timepoints described for Fig. 1a, and then performed chromatin-RNA sequencing (ChromRNA-seq), which gives an enriched fraction of nascent RNAs that has been previously shown to be enriched in eRNAs[41,42]. H3K4me3, a typical active promoter mark, but also present in some active enhancers[43,44] was also determined. The

presence or not of histone marks in the 29,071 putative enhancers determined by ATAC-seq allowed us to identify 4576 primed enhancers (only H3K4me1), 21,794 active enhancers (H3K4me1 and H3K27ac), and 2697 regions with neither H3K4me1 nor H3K27ac (likely regions with levels of marks below our threshold, which we have considered as undefined), prior to TGFβ stimulation in NMuMG cells (see Supplementary Data 2). Since the level of H3K27ac is an excellent indicator of enhancer activation or repression[17,45,46], we used this mark to classify the enhancers upon TGFβ treatment. We found the following categories (Fig. 3a and Supplementary Fig. 5): early-activated (*n* = 2166), late-activated (*n* = 2242), early-repressed (*n* = 1059) late-repressed (*n* = 2654) and TGFβ-independent enhancers (*n* = 20,820). While we also observed transient-activated (*n* = 109) and transient-repressed (*n* = 21) enhancers, we did not analyze these categories further due to too small number of components for statistically significant conclusions. Thus, TGFβ treatment alters the epigenetic marks of about 28% (8251) of enhancers. Of the activated enhancers, 2.3% (102) were latent, 10.5% (462) were primed, and 87.2% (3844) were active (which would imply that H3K27Ac was present and increased upon TGFβ treatment). As expected, 100% of the TGFβ-repressed enhancers were active before TGFβ treatment. Supplementary Data 2 contains information about the status of all identified enhancers from NMuMG cells, before and after TGFβ treatment. The levels of H3K4me1 and H3K4me3 marks followed a pattern similar to H3K27ac, although the changes were smaller and delayed. For example, H3K4me1 levels decreased (Log$_2$FC $\leq -1$) in only 1.5% of the early-repressed enhancers after 2 h of treatment but in 70% of them after 12 h (Fig. 3a and Supplementary Fig. 5), indicating that enhancer decommissioning follows enhancer repression with a slower kinetics.

ChromRNA-seq transcripts associated with a 400 bp window surrounding the ATAC-seq peak (summit ± 200 bp) of intergenic enhancers were considered as eRNA (see examples in Supplementary Fig. 6a). No correlation between the H3K27ac and the eRNAs levels prior to TGFβ treatment was observed (Supplementary Fig. 6b). However, following TGFβ treatment, there was a strong correlation between the FC of H3K27ac and that of eRNA (Supplementary Fig. 6c, d), confirming that change of H3K27ac signal is a good indicator of enhancer activity. Consistently, expression of eRNAs increased in early- and late-activated enhancers and decreased in early- and late-repressed enhancers (Fig. 3b).

We investigated the relationship between enhancer activation/repression (via changes in H3K27ac levels) and changes in chromatin accessibility (via ATAC-seq signal changes). While most of the enhancers of all categories displayed a TGFβ-dependent increase of accessibility (Fig. 3c), those enhancers with a larger increase of accessibility (FC ≥ 4) also displayed a more

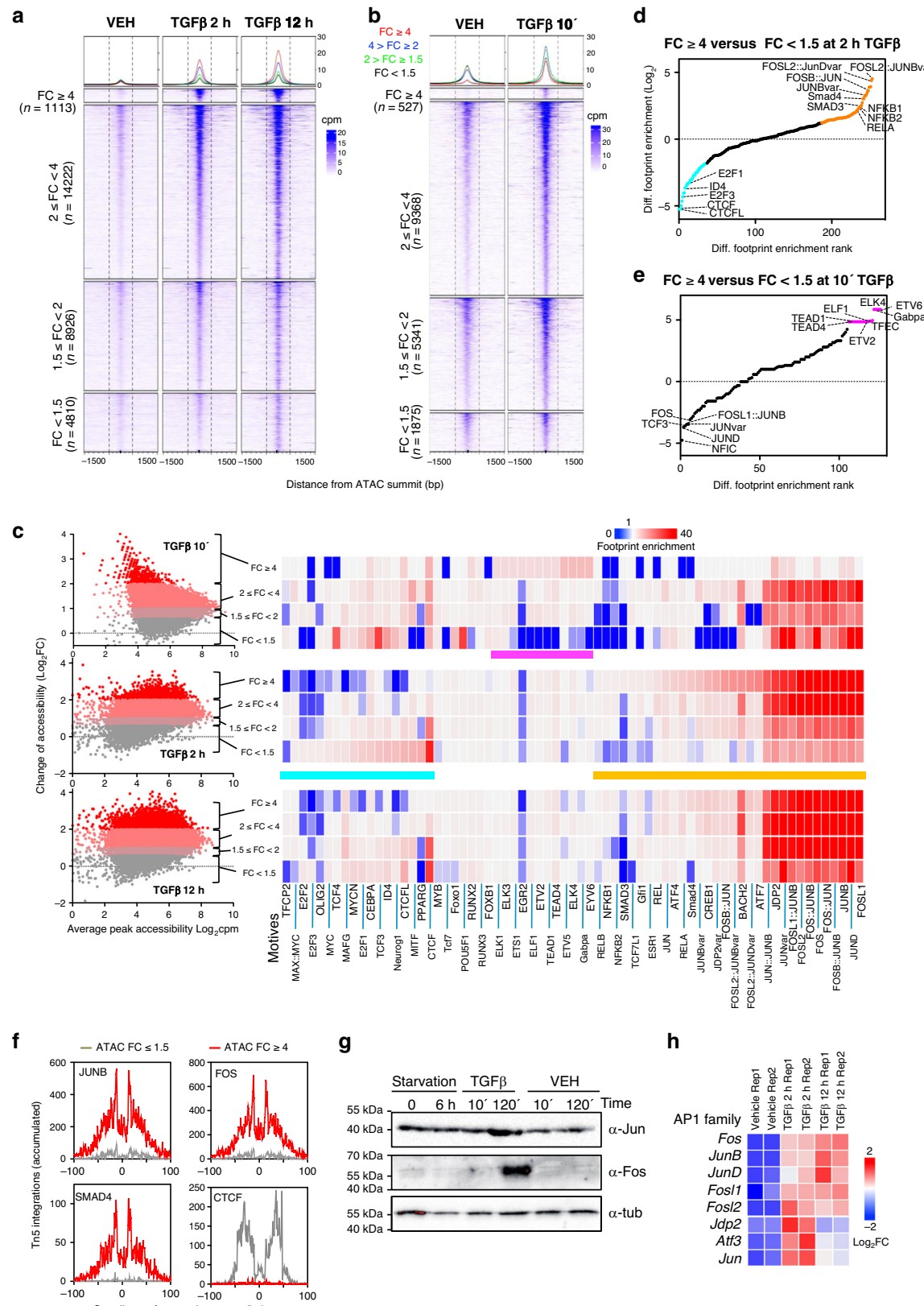

pronounced increase of H3K27 acetylation (Fig. 3d). Examples of increased chromatin accessibility at activated, repressed or TGFβ-independent enhancers are shown in Fig. 3e–g. Thus, widespread chromatin opening occurs irrespective of enhancer activation or repression, with a positive correlation between strong activation and strong increase of accessibility.

Next, we computed enrichment of genomic TF footprints at each category of regulated enhancers. Activated enhancers were enriched in SMAD3, SMAD4 and AP-1 family footprints (Fig. 3h). A strong SMAD2/3 occupancy at early-activated enhancers (Fig. 3i and Supplementary Fig. 7) was also observed by ChIP-seq. In contrast, repressed enhancers were positively

**Fig. 2 ATAC-seq analysis of TGFβ-treated cells. a**, **b** Heatmaps showing alignment of ATAC-seq peaks signal from cells treated with vehicle, or TGFβ for 2 h or 12 h (**a**) or with vehicle or TGFβ for 10 min (**b**). Peaks are divided into four categories (FC < 1.5, 1.5 ≤ FC < 2, 2 ≤ FC < 4 and FC ≥ 4) based in change of ATAC-seq signal between TGFβ and vehicle for 2 h (**a**) or for 10 min (**b**). **c** TF occupancy depending on changes in accessibility at the indicated timepoint. Left: scatter plot of changes of ATAC-seq signal (y-axis, $\log_2$FC) versus average signal (x-axis). ATAC-seq peaks were divided into four categories according to FC as in (**a** and **b**). Right: TF genomic footprints enrichments were determined in the indicated category of ATAC-seq peaks. TF footprints were determined by calculating the protection from transposition observed in the ATAC-seq signal. TF motifs (from JASPAR) were identified in footprinted regions. Seventy of 550 are shown. **d**, **e** Differential footprint enrichment between FC ≥ 4 and FC < 1.5 of TGFβ or vehicle for 2 h (**d**) or for 10 min (**e**). Dots correspond to the same colored bars in **c**. **f** Example of TF footprint profile at 2 h after TGFβ. Accumulated Tn5 integrations (y-axis) along the indicated TF binding sites at nucleotide resolution (x-axis, bp from center of each TF motif) in ATAC peaks with FC ≥ 4 (red) and FC < 1.5 (gray). **g** Western blotting showing levels of Jun and Fos proteins under the indicated conditions. A representative image out of three independent biological replicates is shown. **h** Heatmap showing changes of mRNA levels by RNA-seq ($\log_2$FC) of genes encoding members of the AP-1 family. Source data are provided as a Source Data file.

enriched in TEAD, HNF1A and HNF1B footprints, and depleted of AP-1 family footprints (Fig. 3h). Notably, we found no enrichment of SMAD3 or SMAD4 footprints at repressed enhancers; however, SMAD2/3 binding was detected by ChIP-seq at late-repressed enhancers, suggesting that SMAD2/3 requires interaction with other TFs to bind these repressed enhancers (Fig. 3i and Supplementary Fig. 7b, c).

**Most TGFβ-dependent genes are in the TGFβ-regulated enhancers neighborhood.** We next investigated the location of TGFβ-regulated enhancers with respect to the TGFβ-regulated genes (as determined by RNA-seq). We first labeled genes surrounding a regulated enhancer as Eg (Enhancer gene) ±n, whereby n = 1, 2, 3 or 4, according to gene's upstream (−) or downstream (+) position and chromosome order with respect to the enhancer. For every TGFβ-regulated enhancer, changes of mRNA levels of the neighboring genes (Eg ± 1 to Eg ± 4) were plotted. Genes surrounding an enhancer showed a strong tendency to be regulated in the same way as the enhancer. This effect was not observed when enhancer order positions were randomized (Fig. 4a and Supplementary Fig. 8a). Specifically, about 60% of upregulated and 40% of downregulated TGFβ-dependent genes are in a neighborhood of ±3 genes from an enhancer regulated in the same way ($p = 7.63 \times 10^{-6}$ and $7.05 \times 10^{-14}$, respectively). A significant regulatory effect of the enhancer on the neighboring genes was observed up to about 150 kb upstream or downstream of the enhancers (Fig. 4b and Supplementary Fig. 8b). Approximately 79% of upregulated and 56% of downregulated genes are in a neighborhood of 150 kb from an enhancer regulated in the same way ($p = 9.6 \times 10^{-31}$ and $5.5 \times 10^{-19}$, respectively). Importantly, a strong linear relationship was observed between changes of the enhancer H3K27Ac levels and the change of gene expression of the enhancer's closest gene. This correlation is maintained with the second, third and fourth genes closer to the enhancer (Fig. 4c), albeit with a decreasing regression line slope, indicating that, on average, the enhancer effect is greater at proximal genes and decreases with the distance and/or the presence of genes in between. As enhancers can regulate genes at large distances by looping mechanisms, we wondered whether they operate in a continuous manner or skip over interposed genes. Indeed, enhancers create a continuous influence over their neighborhood: when the second-closest gene (up- or downstream, Eg±2) is robustly regulated ($|\log_2$FC$| >1$; adjusted $p < 0.05$) in the same way as the enhancer, the interposing gene (Eg-1 or Eg+1, respectively) also tends to be regulated that way (Fig. 4d and Supplementary Fig. 9).

**TGFβ-regulated genes are often in co-regulated clusters.** Next we analyzed the genomic distribution of genes regulated by TGFβ. First, genes robustly regulated by TGFβ treatment

(adjusted p-value < 0.05 and $|\log_2$FC$| >1$) were divided into six gene categories: early-upregulated (411), early-downregulated (194), late-upregulated (542), late-downregulated (1129), transient-upregulated (49) and transient-downregulated (36) (Supplementary Fig. 4b and Supplementary Data 1). Altogether, TGFβ promoted robust expression changes in 2361 genes, affecting about 11% of protein-coding genes. As transient categories presented a small number of genes, they were not considered for further analysis. We frequently found several contiguous genes regulated in the same way (Fig. 5a, b). In fact, we found more pairs of co-regulated genes than randomly expected (Supplementary Fig. 10a). To verify this trend, we investigated the transcriptional behavior of genes in the vicinity of a robustly TGFβ-regulated gene by first numbering genes according to their 5′-3′ chromosomal order (1, 2, …, n), and then plotting the FC of genes n ± 1 to n ± 4 for every robustly TGFβ-regulated gene. Indeed, upstream and downstream neighboring genes of a robustly TGFβ-regulated gene (but not of a random gene) tend to be co-regulated (Fig. 5c and Supplementary Fig. 10b). Co-regulation decreased with the number of interposed genes, but significant co-regulation was observed up to the n ± 3 position in early categories, and to the n ± 2 position in late categories. Significant co-regulation was observed for up to about 100 kb upstream and downstream of a robustly TGFβ-regulated gene (Fig. 5d and Supplementary Fig. 10c, d). These data suggest the existence of domains in the genome where genes tend to be co-regulated by TGFβ. Notably, co-regulation also influenced intergenic transcription (Fig. 5e and Supplementary Fig. 10e), and the chromatin landscape (Fig. 5f and Supplementary Fig. 10f, Supplementary Fig. 11). In summary, our co-regulation analyses suggest that TGFβ regulation often operates in clusters of genes.

**Characterization of TGFβ regulatory domains.** The results exposed in the last two sections suggest that TGFβ-dependent enhancers create genomic regions of influence that often expand to several co-regulated genes. Analysis of co-regulation demonstrated that robustly regulated genes are often surrounded by differentially expressed genes with a lower FC. In order to identify most regions of co-regulation, we looked for regions of the genome containing at least two contiguous TGFβ-co-regulated genes, using a less stringent but still significant threshold ($|\log_2$FC$| \geq 0.5$; $p < 0.05$). We found 372 of these clusters (86 early-upregulated, 66 late-upregulated, 49 early-downregulated, 173 late-downregulated) (Supplementary Data 3), which we termed TGFβ regulatory domains (TRDs). TRDs have an average size of 115 kb (with a range from 1 kb to 1.1 Mb) and an average of 2.2 genes (with a range from 2 to 7). About 80% of the upregulated and 53% of the downregulated TRDs contained at least one co-regulated enhancer or had one in its vicinity (±50 kb) ($p = 3.3 \times 10^{-19}$ and $p = 7.7 \times 10^{-10}$, respectively). Meta-analysis of the TRDs demonstrated coordinated regulation of gene mRNA,

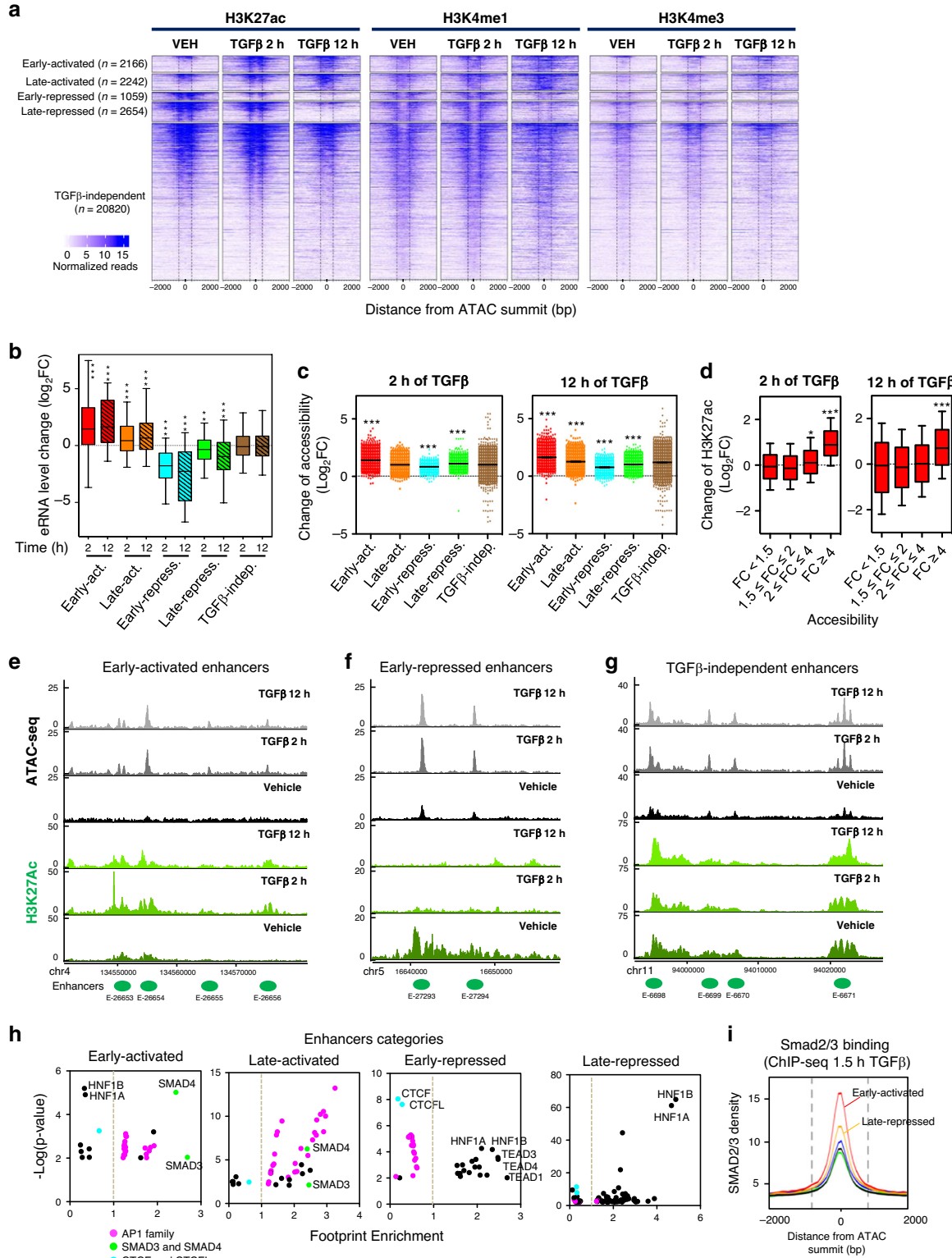

intergenic RNA and H3K27Ac levels as well as a sharp difference of the TRD with respect to the surrounding regions (Fig. 5g). Similar gene ontology (GO) categories were found both in TRD and non-TRD TGFβ−regulated genes (Supplementary Fig. 12a). TRDs tend to be present in regions of high gene density (Supplementary Fig. 12b). Furthermore, the distance between a gene and the closest co-regulated enhancer was shorter in TRD genes than in non-TRD genes. However, there were no differences when distance between a gene and the closest anti-regulated

enhancer were computed (Supplementary Fig. 12c, d). Total gene expression and TGFβ-dependent FCs were similar between TRD and non-TRD genes (Supplementary Fig. 12e, f).

**Enhancers control expression of several genes within a TRD.** We next speculated that co-regulation within a TRD is created by the promiscuous activity of an enhancer. To test this hypothesis, we selected three TRDs (the *Amigo1-Cyb561d1-Atxn712*, *Arid5a-*

**Fig. 3 Identification of distinct classes of enhancers regulated by TGFβ. a** Heatmaps showing ChIP-seq signal of H3K27ac, H3K4me1, and H3K4me3 at distinct categories of enhancers and at the indicated times. **b** Change of eRNA levels of different categories of enhancers shown in (**a**), at the indicated time. Intergenic eRNA-expressing enhancers were selected as described in Methods. Early-activated, $n = 328$; Late-activated, $n = 313$; Early-repressed, $n = 131$; late-repressed $n = 418$; TGFβ−independent, n=2414 enhancers. **c** Change of accessibility (ATAC-seq signal) of different categories of enhancers shown in **a**, 2 h (left panel) and 12 h (right panel) after TGFβ treatment. Each dot corresponds to an enhancer: early-activated ($n = 2166$), late-activated ($n = 2242$), early-repressed ($n = 1059$) late-repressed ($n = 2654$) and TGFβ-independent enhancers (20820). **d** Change of H3K27ac levels, 2 h (left panel) and 12 h (right panel) after TGFβ treatment, of the different enhancers classified according to change of accessibility as in Fig. 2a. FC < 1.5 ($n = 4810$), $1.5 \leq FC < 2$ ($n = 8926$), $2 \leq FC < 4$ ($n = 14222$) and $FC \geq 4$ ($n = 1113$). Statistical significance in **b**–**d** are given with respect to the TGFβ-independent category of enhancers (**b**, **c**) or the category FC < 1.5 (**d**) and were determined with the two-tailed Mann–Whitney non-parametric test, *$p \leq 0.01$; **$p \leq 0.001$; ***$p \leq 0.0001$. Exact p-values are provided in Supplementary Data 5. **b**, **d** The horizontal black line of the boxplot and scatter plot represent the median and the mean value, respectively. The box spans the 25th to 75th percentiles, and whiskers indicate 5th and 95th percentiles. **e**–**g** Screen shot of H3K27ac and ATAC-seq profiles at three different genomic regions. While ATAC signal increased in all enhancers shown, H3K27ac signal increases (**e**), decreases (**f**), or remained unchanged (**g**), upon TGFβ treatment. **h** Enrichment of TF footprints at the indicated categories of enhancers with respect to the rest of categories. On the y-axes, $-\log_{10}$ (p-value) of enrichment (Fisher's exact test) for each TF is represented. **i** SMAD2/3 ChIP-seq signal density in the different categories of enhancers. Source data are provided as a Source Data file.

*Kansl3* and *Lad1-Tmmt2-Pkp1* clusters) and inhibited enhancer activity of one enhancer inside of each cluster by CRISPR interference (CRISPRi)[47] (Fig. 6a, c, e), using two distinct small guide RNAs (sgRNAs) for each enhancer (see Supplementary Fig. 13, for sgRNA targeting). Indeed, enhancer inactivation impaired TGFβ-dependent upregulation of expression of all genes within the TRD (Fig. 6b, d, f). As controls, we showed that neither the TGFβ-non regulated gene *Gnai3* (contiguous to *Amigo1-Cyb561d1-Atxn712)* nor the TGFβ-repressed gene *Neurl3* (contiguous to *Arid5a-Kansl3*) were affected by enhancer inactivation (Supplementary Fig. 14). In summary, our data demonstrate that a single enhancer can control the expression of several co-regulated linked genes within a TRD.

**Relationship between TRD and chromatin 3D organization**. Numerous data indicate that enhancers interact with promoters through a loop mechanism[48]. By analyzing Hi-C data from TGFβ treated NMuMG cells[49], we detected a higher frequency of contacts between TRD enhancers with promoters of the same TRD than with other close promoters (Supplementary Fig. 15a). However, only 0.9% of TRDs correspond to Topologically Associating Domains (TADs) (which are characterized by preferential self-interaction in Hi-C experiments[50]). Further, a strong directionality index[50] change was found at TAD borders (as expected) but not at TRD borders (Supplementary Fig. 15b). The average size of TADs (1 Mb) does not correspond to the average size of TRDs (115 kb) either. Finally, while TRDs are defined by gene co-regulation, most TADs contain genes differently affected by TGFβ (Fig. 7a–d). These data indicate that TRDs are not TADs.

However, we also found a number of TADs with all their expressed genes affected by TGFβ in the same way (93 at 2 h and 70 at 12 h; only TADs with >3 expressed genes were considered) (see example in Fig. 7d). To determine whether this reflected an statistically significant tendency of coordinated regulation within TADs, we calculated the percentage of genes per TAD with a positive or negative FC after TGFβ treatment[51]. We observed more TADs than randomly expected with all their genes activated ($p = 7.7 \times 10^{-6}$ for 2 h and $1.7 \times 10^{-5}$ for 12 h) or repressed ($p = 1.7 \times 10^{-28}$ for 2 h and $1.2 \times 10^{-7}$ for 12 h) by TGFβ, while we observed fewer TADs than randomly expected with around 50% of genes activated or repressed ($p = 5.9 \times 10^{-15}$ for 2 h and $2.5 \times 10^{-7}$ for 12 h) (Fig. 7e and Supplementary Fig. 15c, Supplementary Data 5). Taken together, these results suggest that while TRDs are distinct from TADs, TAD organization influences co-regulation of the genes by TGFβ.

Next, we investigated the role of TAD boundaries on gene-gene and enhancer-gene co-regulation. We observed a clear correlation

between TGFβ-dependent change of mRNA levels of every two contiguous neighboring genes that disappeared when gene positions were randomized (Fig. 7f and Supplementary Fig. 15d). Notably, contiguous genes separated by a TAD boundary had little or no co-regulation (Fig. 7f and Supplementary Fig. 15d), indicating that TAD boundaries are critical for suppressing co-regulation between contiguous genes. Similarly, the strong correlation that we found between H3K27ac and gene expression changes of close enhancer-gene pairs (Fig. 4c) was lost when enhancer-gene pairs separated by a TAD border were analyzed (Fig. 7g and Supplementary Fig. 15e). In contrast, if TAD borders were randomized, the linear correlation was recovered, demonstrating that, as expected, TAD borders insulate TGFβ-regulated enhancers from TGFβ-regulated genes.

Finally, we have computed the occupancy of CTCF at TAD and TRD borders by using mouse mammary ChIP-seq data and our CTCF ATAC-seq footprints. Both methods clearly demonstrated high and very significant density of CTCF at TAD borders (Supplementary Fig. 16), as expected. However, although some increase was observed at TRD borders using ChIP-seq data, it was not significant with respect to the background, suggesting that TRD limits are mostly not determined by CTCF.

**Discussion**
In this study, we reveal that TGFβ causes a fast and widespread increase of accessibility at enhancers and TSS, using both canonical and non-canonical TGFβ pathways. We also describe the enhancer repertoire of NMuMG cells and characterize its regulation by TGFβ, suggesting a role of AP-1 together with SMAD proteins in TGFβ−dependent enhancer regulation. We extensively investigated the genomic organization of TGFβ-regulated enhancers and genes. Most TGFβ-regulated genes are close to TGFβ-regulated enhancers. Often, TGFβ-regulated genes are grouped into what we term TGFβ regulatory domains (TRDs), which differ from TADs but are constrained by TAD boundaries. Enhancers display a promiscuous activity within TRDs, with one enhancer able to regulate several genes.

Signal-driven TFs (SDTFs) can activate specific enhancers, mostly among the repertoire of cell-type-specific accessible enhancers[5,6,10,15], although de novo activation from latent enhancers has been also reported[16,17]. Indeed, we show that most TGFβ-regulated enhancers were already active or primed before TGFβ treatment. Unexpectedly, however, we also found that shortly after addition of TGFβ, a massive chromatin opening occurs at most active and primed enhancers of the genome and in a number of latent enhancers, irrespectively of whether the enhancer will be activated, repressed, or not regulated. This result clearly establishes and separates two different regulatory events:

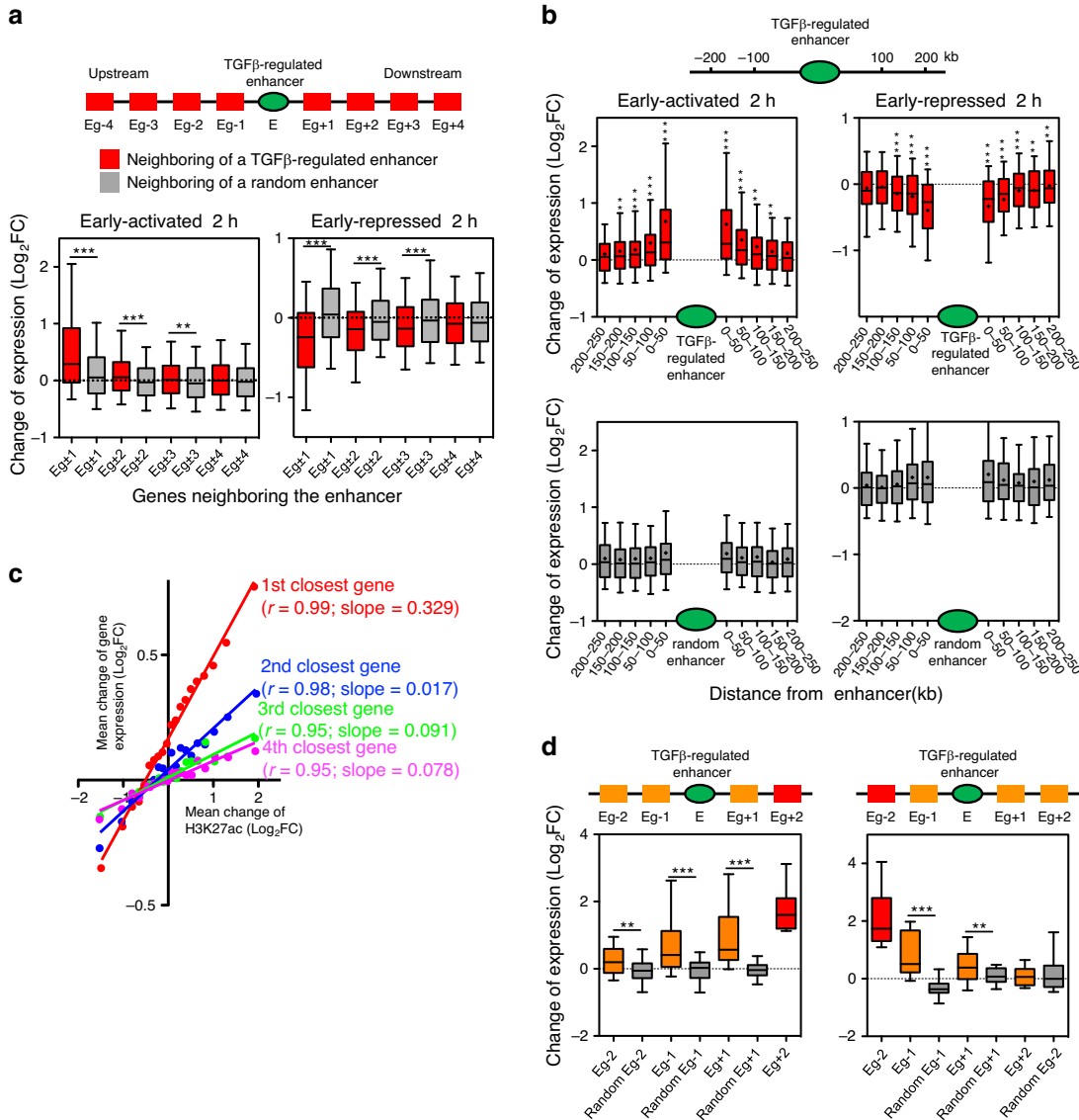

**Fig. 4 Most TGFβ-dependent genes are in the TGFβ-regulated enhancers neighborhood. a** Change of mRNA levels (RNA-seq signal of 2 h TGFβ versus vehicle) for genes upstream or downstream of an early-activated (left) or early-repressed (right) enhancer, as in the scheme (red boxes), or of a randomly selected enhancer (gray boxes). For randomization, see Methods. Eg±n (whereby $n = 1, 2, 3$ or 4) depicts genes that occupy the first, second, third, or fourth positions, respectively, in the chromosomal order, upstream (−) or downstream (+) of the enhancer. Plots for late-enhancer categories are shown in Supplementary Fig. 8a. **b** Change of mRNA levels (RNA-seq signal of 2 h TGFβ versus vehicle) of genes located at the indicated distance (kb) upstream or downstream of an early-activated (top left) or early-repressed (top right) enhancer. Inclusion of the genes in the interval was determined by the position of their TSSs. Lower panels represent the corresponding data when enhancers were randomized. Plots for other enhancer categories are shown in Supplementary Fig. 8b. **c** Correlation plot between changes of mRNA levels (RNA-seq signal of 2 h TGFβ versus vehicle) of the first, second, third, or fourth closest genes respect to an enhancer, versus the change of H3K27ac of the enhancer (ChIP-seq signal of 2 h TGFβ versus vehicle). Data were binned into 20 intervals. Spearman correlation coefficient and slope of the regression line are shown. **d** Change of mRNA levels (RNA-seq signal of 2 h TGFβ versus vehicle) for genes that are located around an early-activated enhancer. Only enhancers for which the second-closest gene (marked in red) upstream (Eg-2) or downstream (Eg+2) was robustly upregulated (log₂ FC > 1; adjusted $p < 0.05$) were considered. Gray color boxes correspond to change of mRNA levels of genes around a random enhancer. Plots for other enhancer categories are shown in Supplementary Fig. 9. **a**, **b**, **d** Statistical significance between real and random distributions were determined with the two-tailed Mann–Whitney test, *$p ≤ 0.05$; **$p ≤ 0.01$; ***$p ≤ 0.001$. Sample size of each distribution and exact $p$-values are provided in Supplementary Data 5. The horizontal black line of the boxplot represents the median value, the box spans the 25th to 75th percentiles, and whiskers indicate 5th and 95th percentiles. Source data are provided as a Source Data file.

the change of chromatin accessibility and the enhancer activation or repression decision. Thus, we speculate that, for some stimuli, the cell first uses an indiscriminate strategy to make most of the active and primed enhancers accessible, prior to moving to a more specific regulation that specifies, which enhancers will be activated or repressed. However, we cannot rule out that pervasive chromatin opening is part of a rapid stress response that

might not be specifically related to TGFβ. More ATAC-seq dynamic studies are required to verify whether widespread opening is a prerequisite for signal-dependent enhancer selection triggered by other stimuli.

Which factors are responsible for this widespread enhancer opening? We showed that silencing SMAD4 impaired widespread chromatin opening. Furthermore, we found that SMAD3/4

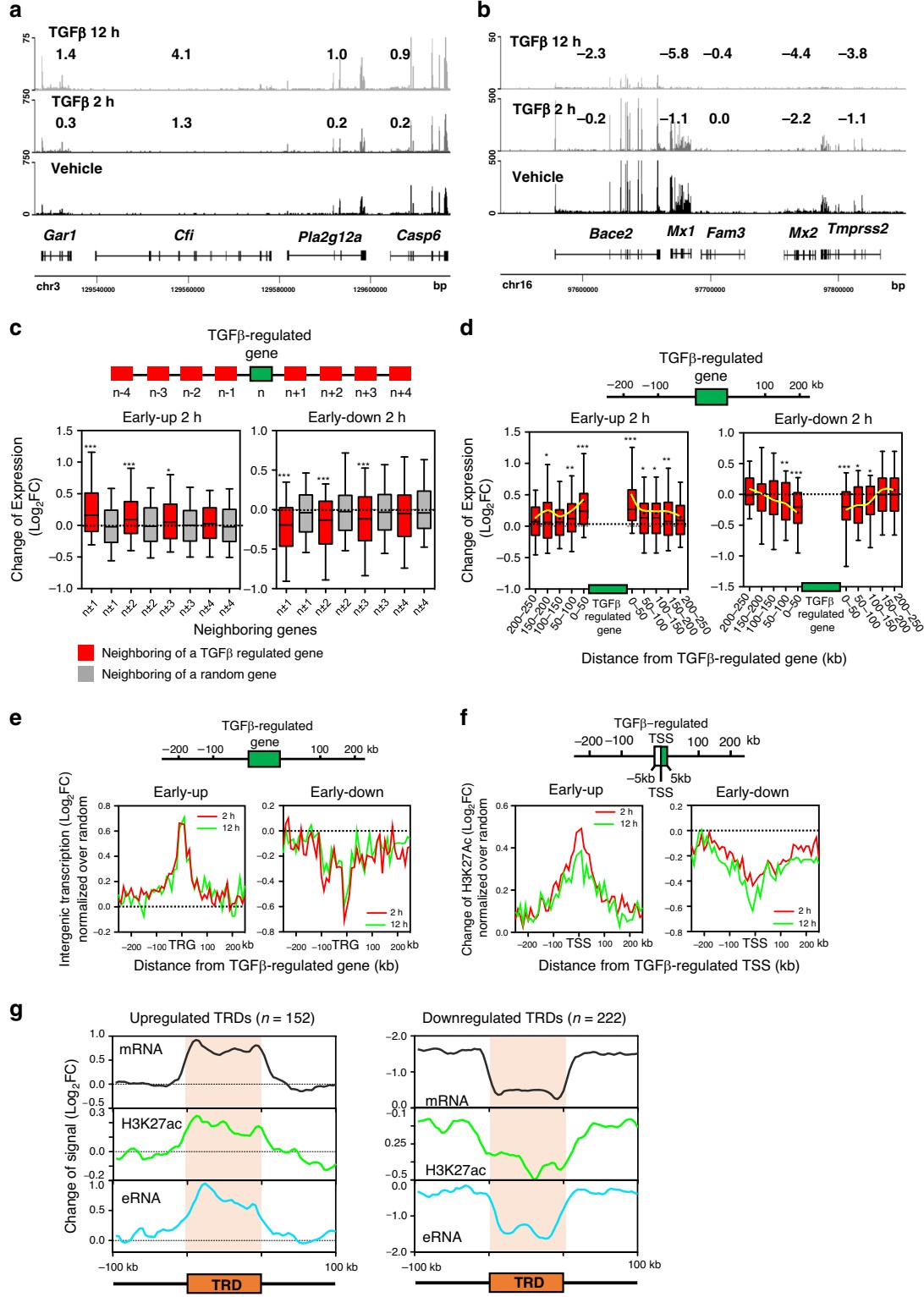

occupancy is enriched in enhancers that strongly gained accessibility at 2 h and 12 h after TGFβ addition. However, SMAD2/3 was only present in about 25% of all enhancers that gain accessibility. In fact, AP-1 footprints were strongly enriched in most enhancers, with a strong correlation between AP-1 occupancy and the increase of enhancer accessibility at 2 h and 12 h after TGFβ addition. Direct interaction and transcriptional cooperation between SMAD proteins and components of AP-1 (such as JUN, JUNB and FOS) have been reported[7,52,53]. Consistently, a

fraction of SMAD2/3-containing enhancers that strongly gained accessibility, also contained footprints for AP-1, suggesting cooperation of these factors for enhancer opening. An important role of AP-1 in enhancer opening and BAF chromatin remodeling complex recruitment has been demonstrated in different cell types[16,54–57].

In addition to the canonical TGFβ pathway, inhibition of kinases of the non-canonical pathway (MEK and TAK1) abolished chromatin opening. MEK activates ERK, a well-known

**Fig. 5 Co-regulated clusters of TGFβ-regulated genes. a, b** Examples of clusters of TGFβ co-regulated genes. Screenshot of RNA-seq tracks at two different genomic regions. Numbers show $Log_2FC$ of the indicated gene at the corresponding timepoint of TGFβ versus vehicle. **c** Changes of mRNA levels (RNA-seq) of the four genes located upstream ($n-1$ to $n-4$) and downstream ($n+1$ to $n+4$) of a robustly TGFβ-regulated gene ($|log_2FC| > 1$; adjusted $p < 0.05$) at position n. The central gene ($n$) is early-upregulated (left) or early-downregulated (right). Gray boxes correspond to the changes of mRNA levels of genes around a random gene. For randomization, see Methods. **d** Change of mRNA levels (RNA-seq) of genes located at the indicated distance (kb) upstream or downstream of a TGFβ-regulated gene. Left: the central gene is early-upregulated. Right: the central gene is early-downregulated. Randomizations are shown in Supplementary Fig. 10d. **c, d** Statistical significance between real and random distributions were determined with the two-tailed Mann–Whitney non-parametric test, $*p \leq 0.05$; $**p \leq 0.01$; $***p \leq 0.001$. Sample size ($n$) of each distribution and exact p-values are provided in Supplementary Data 5. The horizontal black line of the boxplot represents the median value, the box spans the 25th to 75th percentiles, and whiskers indicate 5th and 95th percentiles. A black dot in the boxplot represent the mean. **e** Changes of intergenic transcription in the neighborhood (±250 kb binned in 10 kb bins) of a TGFβ-regulated gene (TRG) normalized to random. ChromRNA-seq data were used. To avoid termination read-through and promoter-divergent transcription, regions of 2 kb upstream and downstream of the TSS and transcription termination site were not considered. **f** Change of H3K27ac (ChIP-seq signal) in the neighborhood (±250 kb binned in 10 kb bins) of a TGFβ-regulated TSS normalized to random. To avoid histone modifications of the TGFβ-regulated TSS, regions of 5 kb upstream and downstream of the TSS were not considered. **c–f** Late-upregulated and late-downregulated categories are shown in Supplementary Fig. 10b, c, e, f. **g** Meta-analysis of TRDs. Density plots of changes of mRNA, eRNA, and H3K27ac levels are shown. Data from early- and late-upregulated or downregulated TRDs were pooled. Source data are provided as a Source Data file.

player of the TGFβ signaling[58]. ERK may regulate chromatin opening in several ways, including AP-1 and other transcription factors regulation[59–63], phosphorylation of histone H3 at serine 10 through MSK1[64,65] and direct regulation of the chromatin and transcription machinery[66,67]. TAK1 is also a component of the TGFβ non-canonical signaling pathway that acts upstream of JNK[58,68], which regulates AP-1. Therefore, our data point to both SMAD2/3/4 and AP-1 TFs as being mainly responsible for pervasive enhancer chromatin opening at 2 h and 12 h after TGFβ. However, AP-1 and SMAD2/3/4 footprints were not enriched in highly opened chromatin after 10 min. Furthermore, at this early time, Jun and Fos protein levels have not yet been induced by TGFβ. At this time ETS TF footprints were enriched in enhancers that strongly gained accessibility. As many ETS factors are direct targets of ERK[69], the ERK-ETS pathway might lead a first wave of accessibility, at least in a subset of enhancers.

Even though about 80% of enhancers in a cell gained accessibility upon TGFβ addition, only about 28% of them presented increased or decreased H3K27ac levels (indicating activation or repression, respectively, by TGFβ) Activated enhancers were strongly enriched in SMAD3, SMAD4, and AP-1 footprints and displayed a stronger chromatin opening. In sharp contrast, repressed enhancers were not enriched in SMAD3 or SMAD4 footprints and were negatively enriched in AP-1 footprints as compared to other enhancers. However, ChIP-seq data demonstrate an important SMAD2/3 occupancy of a number of repressed enhancers. Early-repressed enhancers presented a significant enrichment in footprints of TEAD factors. In human embryonic stem cells, a complex of TEAD factors, SMAD2 and SMAD4 has been shown to repress mesodermal genes through binding and repression of enhancers[70]. Consistent with a role of TEAD factors in TGFβ-dependent enhancer repression also in mammary epithelium, we found upregulation of several TEAD factors (and especially of TEAD2) at 2 h and 12 h of TGFβ treatment (1.6- and 4-fold upregulation, respectively; Supplementary Data 1). SNAI1 is one of the master regulators of the EMT process[71] and the *Snai1* gene is strongly upregulated by the TGFβ treatment (see Supplementary Data 1). Notably, we did not find SNAI1 motifs or footprints in the enhancers identified in this work. Several works have shown binding of SNAI1 to promoters[72–79], suggesting that this factor is more likely associated with promoters rather than with enhancers.

Enhancers can regulate genes from large distances through genomic loops[48]. Our data indicate, however, that most genes robustly regulated by TGFβ (79% of the upregulated, and 56% of the downregulated) were in the close vicinity of an enhancer that is also regulated by TGFβ in the same direction. This tendency for

parallel regulation often extended to several genes around the enhancer in a continuous manner (e.g., jumping of interposed genes is not frequent), indicating that enhancers generate regions of gene co-regulation. Indeed, we found 372 clusters of co-regulated genes, which we termed TRDs. Most TRDs contain at least one enhancer within their limits or in their close proximity, and their average size is about 115 kb. Previous evidences of regulatory domains somewhat similar to TRDs have been reported. For instance, after analyzing the reporter activity of hundreds of transposon integrations in the mouse genome, Symmons and co-workers[80] identified 46 regions of the genome (median size of 359 kb) at which adjacent integrations presented similar expression patterns, suggesting that enhancers act in a largely undiscriminating manner within these regions. Based on cohesins ChIA-PET data, Young and collaborators created the concept of insulated neighborhoods as specific DNA-loop structures, including one enhancer and one or two genes regulated by that enhancer. Insulated neighborhoods promote enhancer-promoter interactions, are delimited by CTCF-binding sites and are within TADs but are different from TADs[81–83]. Although some TRD characteristics are consistent with those of insulated neighborhoods, we did not observe a high density of CTCF at TRD borders. Within TRDs, not all genes are regulated with the same intensity by the enhancer. While we found a strong dependence of enhancer-gene proximity (Fig. 4c), there are also multiple cases in which genes at positions ±2 with respect to the enhancer were regulated by TGFβ more strongly than the interposed gene at ±1, indicating that the strength of the enhancer-promoter influence depends not only on the distance and gene order with respect to the enhancer, but also on other promoter-enhancer specificity factors.

Clustering of co-regulated or co-expressed genes has been previously reported[84–87], and enhancers have been proposed as probable regulatory elements responsible for co-regulation[88–90]. However, to our knowledge, functional experiments that demonstrate this statement have not been reported. We show that inactivation of three different enhancers from three TRDs impairs TGFβ-dependent regulation of the two to three genes in the neighborhood of the enhancers, demonstrating that enhancers have a more unrestrained activity than previously anticipated. We favor the view that, at least a subset of TGFβ-dependent enhancers display a rather promiscuous preference for surrounding genes.

To which extent TADs control transcription is still open to discussion[91–93]. Our data indicate that TRDs differ from TADs, and that TADs are not units of co-regulation. Indeed, most TADs contain genes that are differently affected by TGFβ. However, our analysis of the frequency of co-regulated genes in TADs showed a

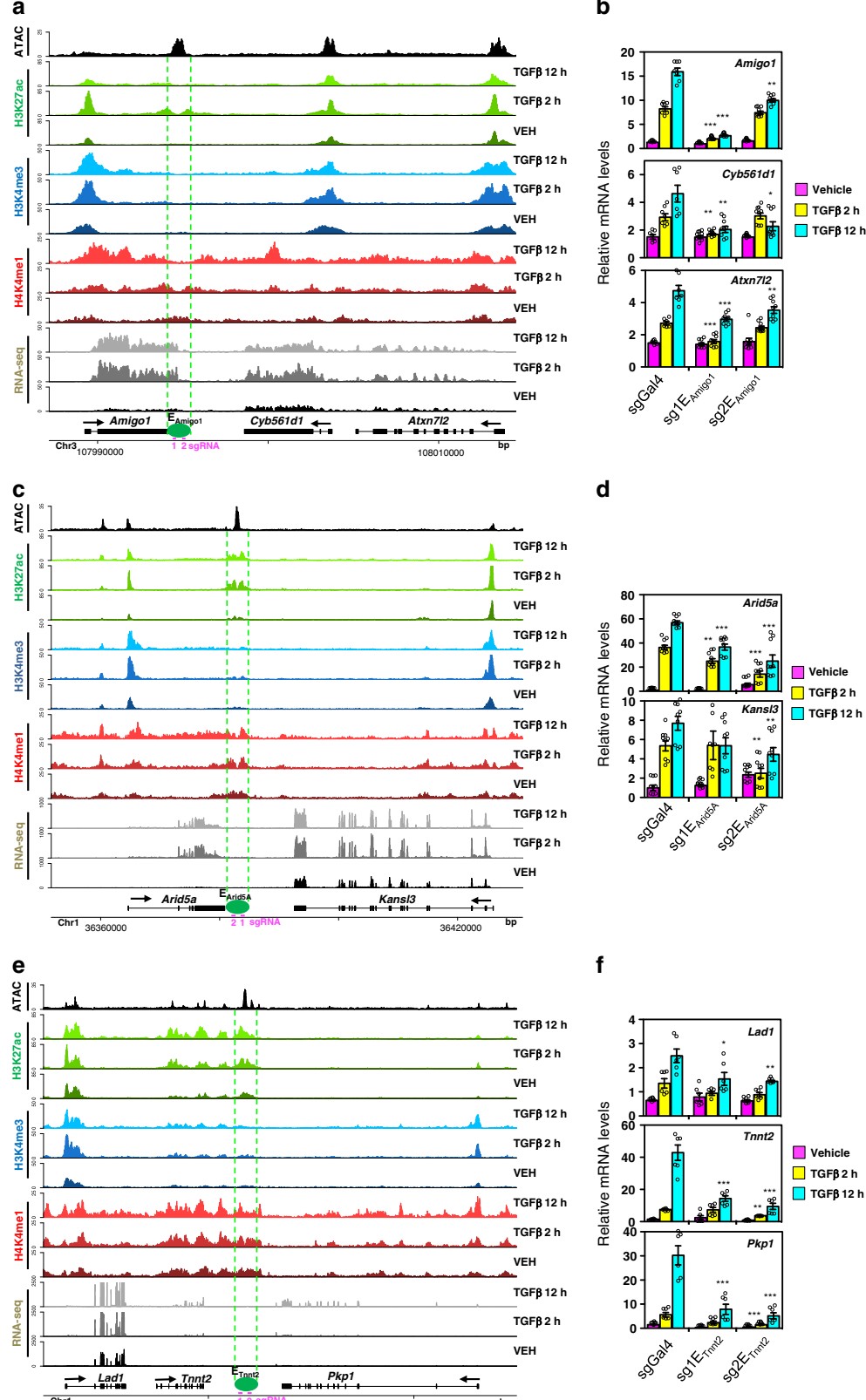

small but significant tendency of the genes of a TAD to be co-regulated, as also reported by others[51,94,95]. It is well-established that TAD borders are enriched in CTCF-binding sites and have enhancer-promoter insulation activity[50]. We confirmed in our system that gene co-regulation and enhancer-gene regulation drastically decrease across TAD borders. Extending the concept

to other regulatory domains (RDs), we favor a model in which RDs are the building block of co-regulation. However, the insulating activity of TAD borders obligates most RDs to reside within single TADs, resulting in a bias towards co-regulation of the genes within TADs (see model in Fig. 7h and real examples in Fig. 7a–c). As RDs have 2 to 3 genes on average, 100% co-

**Fig. 6 CRISPR-mediated enhancer inactivation demonstrates promiscuous activity of enhancers on neighboring genes. a, c, e** Screenshot of H3K27ac, H3K4me1, H3K4me3, RNA-seq, and ATAC-seq tracks at three different genomic regions. Position of a TGFβ-regulated enhancer is shown. For the inactivation of the enhancers by CRISPRi, two different sgRNAs (pink) that target each enhancer were used. **b, d, f** Determination of mRNA levels of the indicated genes by RT-qPCR analysis. Values were normalized to the *Gapdh* mRNA level. Values are average ± SEM of at least six determinations from three independent experiments. Statistical significance of the values with respect to the same timepoint of the negative control (a non-targeting sgRNA, sgGal4) were determined with the two-tailed Mann–Whitney non-parametric test. *$p \leq 0.05$; **$p \leq 0.01$; ***$p \leq 0.001$. Sample size (*n*) of each distribution and exact *p*-values are provided in Supplementary Data 5. Source data are provided as a Source Data file.

regulation of the genes of a TAD is mostly observed in small TADs that only harbor a single TRD. We think that this view can reconcile opposing results describing fundamental or marginal roles of TADs in gene expression[96].

In summary, our work provides a valuable catalog of enhancers and their regulation by TGFβ in the model cell line NMuMG. In addition, our results also shed light about fundamental aspects of genome organization and enhancer biology.

## Methods

**Cell culture and treatments**. Normal murine mammary gland NMuMG cells (provided by José Antonio Pintor-Toro, CABIMER, in 2014) were cultured in Dulbecco's modified Eagle's medium containing 10% fetal bovine serum (FBS) and 10 µg/ml insulin (complete medium), until reaching 70–80% confluency. HEK293T cells were grown in Dulbecco's modified Eagle medium (DMEM) with 10% fetal bovine serum (FBS). Human epithelial MCF7 cells were cultured in RPMI medium containing 10% FBS and supplemented with L-glutamine 2 mM. RPE1 cells were grown in DMEM-F12 medium containing 10% FBS and supplemented with L-glutamine 2 mM. All the cell culture media included penicillin (100 U/ml) and streptomycin (100 µg/ml). TGFβ treatments were performed after 6 h of serum starvation, except when indicated. All serum starvation treatments of the NMuMG cells also include deprivation of insulin. In some experiments, TGFβ treatments were performed in normal complete medium (Supplementary Fig. 2b–e) or after 24 or 48 h of serum and insulin starvation (Supplementary Fig. 2f). For TGFβ treatments, 5 ng/ml TGFβ1 diluted in 4 mM HCl, 1 mg/ml BSA (240-B, R&D Systems) or 4 mM HCl 1 mg/ml BSA (vehicle, as control), was added to the medium for the indicated time.

**ATAC-sequencing**. The ATAC experiments were performed according to ref. [28]. NMuMG cells were treated with either vehicle or TGFβ for 2 or 12 h (after 6 h of serum and insulin starvation), vehicle or TGFβ-treated for 10 min (after 6 h of serum and insulin starvation) or vehicle or TGFβ-treated for 2 h (in complete medium); and then collected and treated with transposase Tn5 (Nextera DNA Library Preparation Kit, Illumina). DNA was purified using MinElute PCR Purification Kit (Qiagen). All samples were then amplified by PCR using NEBNextHigh-Fidelity 2× PCR Master Mix (New Englands Labs) with primers containing a barcode to generate libraries. DNA was again purified using MinElute PCR Purification kit and samples were sequenced using Illumina NextSeq 500 system (Illumina) with 75 bp paired-end reads at the Genomic Unit of CABIMER (Sevilla, Spain)

**ATAC-see experiments**. For ATAC-see experiments, cells were grown on coverslips and processed according to ref. [29]. For kinetic analyses, NMuMG cells were changed to serum- and insulin-free medium for 6 h and then were treated with vehicle or TGFβ for 5, 10 min, 1, 2, and 12 h before fixation with 4% paraformaldehyde in phosphate-buffered saline (PBS) for 10 min. For the study of the canonical pathway, cells were transfected with siCT or DsiSMAD4 (IDT, Leuven, Belgium) (previously used in ref. [97]), using Lipofectamine RNAiMAX (#13778-150, Thermo Fisher) according to manufacturer instructions. After 48 h, cells were serum- and insulin-depleted for 6 h and treated with vehicle or TGFβ for 10 min before fixation. For the study of the non-canonical pathway, cells were first serum- and insulin-depleted for 4 h. Then, cells were treated with DMSO or SB431542 (S4317, Sigma-Aldrich), U0126 (S1102, Selleckchem) and TAKINIB (SML2216, Sigma-Aldrich) inhibitors at 10 µM for 2 additional hours and finally cells were treated with vehicle or TGFβ for 10 min. Other ATAC-see experiments were performed under the conditions detailed in figure legends. After fixation, cells were permeabilized with lysis buffer (10 mM Tris–HCl, pH 7.4, 10 mM NaCl, 3 mM MgCl₂, 0.01% Igepal CA-630) for 10 min at room temperature. Next, coverslips were rinsed in PBS twice and put in a humidity chamber box at 37 °C. The transposase mixture solution (25 µl 2× TD buffer, final concentration of 100 nM Tn5-ATTO-59ON, adding dH₂O up to 50 µl) was added on the coverslips and incubated for 30 min at 37 °C. As a negative control 50 mM EDTA was added to the Tn5-ATTO590N reaction buffer. After the transposase reaction, coverslips were washed with PBS containing 0.01% sodium dodecyl sulfate (SDS) and 50 mM EDTA for 15 min three times at 55 °C. Then, slides were stained with DAPI for 5 min and mounted using Vectashield for imaging.

**DNaseI-seq**. DNase I hypersensitivity mapping was performed according to ref. [98] with brief modifications. NMuMG cells were either vehicle or TGFβ-treated for 2 h. The cells were trypsinized and pelleted before washing and resuspension in buffer A (15 mM Tris-Cl (pH 8.0), 15 mM NaCl, 60 mM KCl, 1 mM EDTA (pH 8.0), 0.5 mM EGTA (pH 8.0), 0.5 mM spermidine, 0.15 mM spermine). Nuclei were extracted by adding buffer A containing NP-40. The nuclei were washed with buffer A and resuspended in pre-warmed lysis buffer at a concentration of 5 × 10⁶/ml and then digested with 75 units of DNase I (Roche) for 5 min at 37 °C. The reactions were terminated by the addition of an equal volume of stop buffer and incubated at 55 °C. After 15 min, proteinase K (final concentration of 20 µg/ml) was added to each digestion reaction and incubated for 16 h at 55 °C. DNA was extracted by careful phenol-chloroform purification using phase-lock gel. DNA fragments of 50–300 bp were selected using low-melting agarose gel and was purified using GenElute Gel Extraction Kit (Sigma-Aldrich). DNA was again purified using MinElute PCR Purification kit and samples were sequenced using Illumina NextSeq 500 system (Illumina) with 75 bp paired-end reads at the Genomic Unit of CABIMER (Sevilla, Spain).

**RNA sequencing**. Total RNA from NMuMG cells, treated with either vehicle or TGFβ for 2 or 12 h, was extracted using the RNeasy kit (74106, QIAGEN). Then, libraries were prepared with the TruSeq Stranded TOTAL RNA kit (Illumina). The sequencing was performed with a HiSeq 2000 system (Illumina) with 50 bp single-end reads at the Genomics *Core* Facility (EMBL Heidelberg).

**ChromRNA sequencing**. Cellular fractionation was carried out according to ref. [99], based on the protocol described in ref. [100]. Cell pellets were resuspended in 400 µl cold cytoplasmic lysis buffer (0.15% NP-40, 10 mM Tris pH 7.5, 150 mM NaCl) and incubated on ice for 5 min. The lysates were layered onto 1 ml cold sucrose buffer (10 mM Tris pH 7.5, 150 mM NaCl, 24% sucrose w/v), and centrifuged in microfuge tubes at 3500 g for 10 min. The nuclear pellets were gently resuspended into 250 µl cold glycerol buffer (20 mM Tris pH 7.9, 75 mM NaCl, 0.5 mM EDTA, 50% glycerol). An additional 250 µl of cold nuclei lysis buffer (20 mM HEPES pH 7.6, 7.5 mM MgCl₂, 0.2 mM EDTA, 0.3 M NaCl, 1 M urea, 1% NP-40, 1 mM DTT) was added to the samples, followed by a pulsed vortexing and incubation on ice for 2 min. Samples were then spun in microfuge tubes for 2 min at 13,000 × *g*. Fifty microliters of cold PBS was added to the remaining chromatin pellet, and gently pipetted up and down over the pellet, followed by a brief vortex. Then RNA was extracted using TRIZOL and treated with TURBO DNA-free kit (AM1907, Invitrogen) for DNA removal according to manufacturer's instructions. Libraries preparation and sequencing was as described in the RNA sequencing section.

**ChIP sequencing**. ChIP assays were performed according to ref. [101] with some modifications. Briefly, NMuMG cells treated with either vehicle for 2 h or TGFβ for 2 or 12 h were crosslinked in 1% formaldehyde for 10 min at room temperature followed by addition of glycine (125 mM final concentration) for 5 min. Nuclei were isolated using lysis buffer 1 (5 mM Pipes pH 8, 85 mM KCl, 0.5% NP-40, and complete protease inhibitor cocktail (Roche)) and were lysed using lysis buffer 2 (1% SDS, 10 mM EDTA, 50 mM Tris–HCl pH 8.1, and complete protease inhibitor cocktail (Roche)). Chromatin was sheared into an average size of 500 bp by eight pulses of 30 s (30 s pause between pulses) at 4 °C in the water bath sonicator Bioruptor (Diagenode, Liège, Belgium). Thirty micrograms of chromatin were diluted 1:10 in IP buffer (0.01% SDS, 1.1% Triton X-100, 1.2 mM EDTA, 16.7 mM Tris–HCl pH 8.1, 167 mM NaCl, 1% sodium deoxycholate) and incubated overnight at 4 °C in rotation with 3 µg of the respective antibodies: anti-H3K27ac (Ab4729, Abcam), anti-H3K4me1 (Ab8895, Abcam) and anti-K3m4me3 (Ab8580, Abcam). Immunoprecipitations were incubated for 2 h at 4 °C in rotation with protein A Dynabeads (Invitrogen) and then washed with wash buffer 1 (0.1% SDS, 1% Triton X-100, 2 mM EDTA, 20 mM Tris–HCl pH 8.1, and 150 mM NaCl, 1% sodium deoxycholate), wash buffer 2 (0.1% SDS, 1% Triton X-100, 2 mM EDTA, 20 mM Tris–HCl pH 8.1, 500 mM NaCl, 1% sodium deoxycholate), wash buffer 3 (0.25 M LiCl, 1% NP-40, 1% sodium deoxycholate, 1 mM EDTA, 10 mM Tris–HCl pH 8.1), and twice with TE1x buffer (10 mM Tris–HCl pH 8.0, 1 mM EDTA). The complexes were eluted twice from the beads with elution buffer (1% SDS in TE1x buffer) by incubating 10 min at 65 °C. The eluates and the inputs were incubated overnight at 65 °C for de-crosslinking and treated with Proteinase K for 1 h at 37 °C. Chipped DNA was purified with phenol:chloroform extraction followed by ethanol precipitation and resuspended in miliQ water. The sequencing was

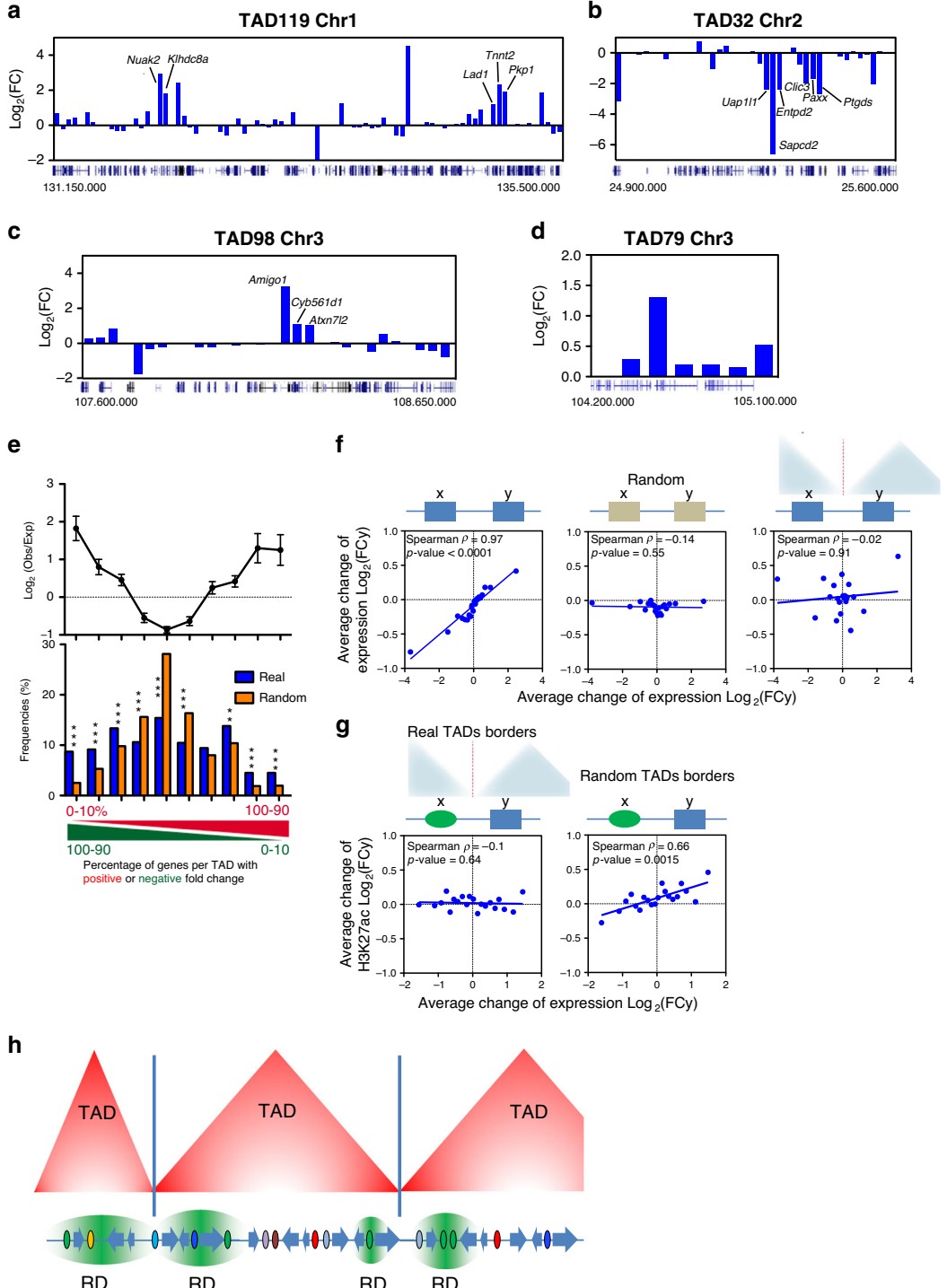

**Fig. 7 TRDs Differ from TADs but are Constrained by TAD Borders. a–d** Changes of mRNA levels (2 h TGFβ versus vehicle) of genes contained at the indicated TADs. Co-regulated genes at TRDs are shown. **e** Top panel: ratio of observed versus expected frequencies of TADs with distinct proportions of genes with upregulated or downregulated FC (FC > 1 or FC < 1; 2 h TGFβ versus vehicle). Values are means ± SD. Bottom: histogram of the frequencies of TADs for the observed (blue) or randomized (orange) position of genes. TADs (n = 688) were binned into ten intervals depending on the percentage of up- versus downregulated genes. Significance was determined by comparing the real value with 500 randomizations of the gene order (see Methods). Probabilities (p) of the real number considering Normal distribution are provided. **p ≤ 0.001; ***p ≤ 0.0001. Exact p-values are provided in Supplementary Data 5. Data for 12 h are given in Supplementary Fig. 15 c. **f** Correlation between change of expression (12 h TGFβ versus vehicle) of every pair of expressed contiguous genes (x, y) of the genome using real chromosomal order (left), random gene order (middle) or pairs of contiguous genes separated by a TAD border (right). Spearman correlation coefficient and p-values are shown. Data were binned into 20 intervals. Data for 2 h are given in Supplementary Fig. 15d. **g** Correlation plot between change of H3K27ac signals of enhancers (ChIP-seq signal 2 h TGFβ versus vehicle) and change of mRNA level (RNA-seq signal 2 h TGFβ versus vehicle) of their closest gene separated by real (left), or by random TAD borders (right). Data were binned into 20 intervals. Data for 12 h are given in Supplementary Fig. 15e. **h** Model of influence of TGFβ-regulatory domains (TRD) and other regulatory domains (RD) constrained by the insulating activity of TAD borders. Small TADs can harbor a single RD. Source data are provided as a Source Data file.

performed with the HiSeq 2000 system (Illumina) with 50 bp single-end reads at the Genomics *Core* Facility (EMBL Heidelberg).

**ChIP qPCR**. ChIP assays were performed as described above for ChIP sequencing, using anti-H3K27ac (Ab4729, Abcam) and anti-CRIPSR-Cas9 (C15310258-100, Diagenode). Rabbit IgG (Sigma) was used as a control for non-specific interactions. Chipped DNA was purified using ChIP DNA Clean & Concentrator kit (Zymo Research), and eluted in 50 μl of elution buffer. Input was prepared with 10% of the chromatin material used for immunoprecipation. Input material was diluted 1:5 before PCR amplification. Quantification of immunoprecipitated DNA was performed by real-time PCR (qPCR) with the Applied Biosystems 7500 FAST real-time PCR system, using Applied Biosystems Power SYBR green master mix. Sequences of all oligonucleotides are listed on Supplementary Data 4. Data are the average of two or three biological independent replicates and three technical replicates of each biological replicate.

**Western blotting**. Western blotting were performed according to ref. [102], using the following antibodies: anti-cFos (2250, Cell Signaling. Dilution 1/1000), anti-cJun (60A8; 9165, Cell Signaling. Dilution 1/1000), anti-alfa-Tubulin (DM1A; T9026, Sigma. Dilution 1/1000), anti-Erk1/2 (P28482, Millipore. Dilution 1/1000), and anti-Phospho-Erk1/2 (thr202/Tyr204) (9101, Cell Signaling. Dilution 1/1000), anti-SMAD4 (SC-7996, Santa Cruz. Dilution 1/250). As secondary antibodies anti-mouse-HRP and anti-rabbit-HRP (Sigma-Aldrich. Dilution 1/1000) were used.

**CRISPR interference**. For enhancer inactivation we used CRISPRi technology described in ref. [47]. First the NMuMG-dCAS9 cell line was established as follows. For lentiviral particles preparation $2 \times 10^6$ HEK293T cells were transfected with Fugene 6 (Promega) using 7.5 μg of the transfer vector pHR-SFFV-dCas9-BFP-KRAB (this plasmid was a gift from S. Qi & J. Weissman (Addgene plasmid # 46911; http://n2t.net/addgene:46911; RRID:Addgene_46911)[47] with 5 and 2.5 μg of the packaging plasmids pCMVDR8.91 and pVSVG, respectively. Lentiviruses were harvested 72 h post-transfection, passed through a 0.45-μm filter and concentrated by ultra-centrifugation at $100,000 \times g$ for 90 min. Virus particles were resuspended in DMEM, snap frozen in liquid nitrogen and stored at −80 °C. After tittering, NMuMG cells were infected with pHR-SFFV-dCas9-BFP-KRAB lentiviruses using polybrene 8 μg/ml. Single BFP-positive cells were sorted 72 h after infection into individual wells of a 96-well plate, containing a mixture of fresh and conditioned medium (1:1), and then one clone was chosen after BFP signal and EMT functional validation. Then, NMuMG cell pools expressing specific sgRNAs were established as follows. To design sgRNAs we selected neighboring region for ATAC-seq peak submit and use IDT (https://eu.idtdna.com/site/order/designtool/index/CRISPR_CUSTOM) and CRIS-POR (http://crispor.tefor.net/) tools. Designed sgRNA for selected enhancers (Supplementary Data 4) were inserted into sgRNA expressing lentiviral vectors pU6-sgGAL4-1 (this plasmid was a gift from S. Qi & J. Weissman (Addgene plasmid # 46915; http://n2t.net/addgene:46915; RRID:Addgene_46915)[47]. Then lentiviral particles were produced and purified as above. After tittering, NMuMG-dCAS9 cells were infected with sgRNA expressing lentiviruses and selected with puromycin 1 μg/mL for one week before working with the selected pool.

**Determination of mRNA by quantitative reverse transcription PCR (RT-qPCR)**. Total RNA was prepared by using the RNeasy Kit (Qiagen), as described in the manufacturer's instructions, including DNase I digestion to avoid potential DNA contamination. Complementary DNA (cDNA) was generated from 3 μg of total RNA using SuperScript First Strand Synthesis System (Invitrogen). Two 2 μl of generated cDNA solution was used as a template for real-time PCR (qPCR). Gene products were quantified by qPCR with the Applied Biosystems 7500 FAST Real-Time PCR System, using Applied Biosystems Power SYBR Green Master Mix. Values were normalized to the expression of the *Gapdh* housekeeping gene. At least three biological replicates and three technical determinations were performed in each case. All oligonucleotide sequences used are listed in Supplementary Data 4.

**Computational methods and statistical analysis**. Most of analyses were performed using R (v3.4.4), Rstudio (v0.99.879) and Bioconductor (v2.38).

**Statement on the use of the mm9 assembly**. We used the GRCm37 (mm9) assembly to map all sequencing reads from mouse origin in this study (ATAC-seq, RNA-seq, ChromRNA-seq, DNaseI-seq and ChIP-seq) because many programs did not support yet the mm10 build when we started our study. As our study compares samples across different treatments and does not perform absolute analyses, re-aligning the reads to GRCm38 (mm10) should not significantly affect our conclusions.

**Processing of ATAC-seq data**. ATAC-seq data were processed (trimmed, aligned, filtered, and quality controlled) for each condition independently, using the ATAC-seq pipeline from the Kundaje lab (https://github.com/ENCODE-DCC/atac-seq-pipeline). Information about read depth can be found in the reporting summary. The model-based analysis of ChIP-seq macs2 (v2.1.2) was used to identify the peak regions with options -B, -q 0.01 –nomodel, -f BAM, -g mm9. The Irreproducible

Discovery Rate (IDR) method was used to identify reproducible peaks between two biological replicates. Only peaks reproducible between the two biological replicates (with IDR ≤ 0.05) were retained for downstream analyses. Peaks for all conditions of each experiment were then merged together into a standard peak list. For the experiment described in Figs. 1a and 2a, 52,626, 100,459, and 105,334 peaks were identified for vehicle, TGFβ2h and TGFβ12h conditions, respectively. Then sets of peaks for each condition were refined. All peaks smaller than 150 bp were discarded, and all peaks closer than 1000 bp were merged together using csaw (v1.12.0)[103], obtaining 16,769, 32,545, and 33,077 peaks for each condition. Then, peaks for the three different conditions closer than 1000 bp were combined in a single set of peaks obtaining 39,432 unique regions. Similar procedures were used for the experiment described in Fig. 2b and Supplementary Fig. 2c–e. In all the cases summits were re-calculated using MACS2 refinepeak with default options. Peaks coordinates for downstream analysis were calculated as summit ± 500 bp. For enhancer assignment, those peaks that overlap with TSS ± 1000 bp of any transcript from mm9 UCSC KnownGene, were discarded using bedtools (v.2.27.1) subtract with -A options, obtaining 29,071 peaks, which were considered as putative enhancers. Correlation between replicates were calculated using multi-BamSummary and plotCorrelation from deeptools (v3.1.3) using default options. Shushi package (v.1.16.0) was used for data visualization.

**Processing of DNaseI-seq data**. DNaseI-seq data was processed following the same pipeline as ATAC-seq data but using specific options for DnaseI-seq. Information about read depth can be found in the reporting summary.

**Classification of enhancers according to increase of accessibility**. To classify enhancers according to the increase of accessibility, reads were counted in full-length ATAC-seq peaks using regionCounts() from csaw with minq = 40 option. To normalize data, first we count reads in full genome using windowCounts() with bin = TRUE, width = 2000 and minq = 40 options; then normalization factors were calculated using normOffsets(). After that, estimateDisp(), glmQLFit() and glmLRT() from edgeR package (v3.20.9) was used for calculating $\log_2$FC and statistics. Enhancers were then classified into four groups: FC < 1.5, 1.5 ≤ FC < 2, 2 ≤ FC < 4 and FC ≥ 4, obtaining 4810, 8926, 14,222, and 1113 for TGFβ2h vs veh comparison; and 3988, 7010, 15,362, and 2771 for TGFβ12h vs. veh. EnrichedHeatmap (v1.12.0) package was used for drawing heatmaps. In Fig. 2a rows were ordered according to decreasing ATAC-seq cpm for TGFβ2h or TGFβ12h (merged replicates). Differential ATAC-seq signal analysis (Supplementary Fig. 1b) was also performed in full-length ATAC peaks overlapping TSS (promoters), putative enhancers and the whole genome outside ATAC-seq peaks (rest) using region-Counts() with same parameters as above.

**Analysis of ATAC-see using TANGO**. Confocal microscopy images were analyzed using TANGO[30] plugin from imageJ[104]. To determine nuclei area we used standard protocol for segmentation and post-filters obtaining 203 nuclei for vehicle condition and 194 for TGFβ2h from two independent experiments. To determine open-chromatin domains we used as pre-filters standard fast filters 3D and LoG 3D (BIG) filter. Then 3D spot detector was used for segmentation and region adjustment and merged regions were finally used as post-filters. Standard measure geometrical simple and signal quantification were performed to analyze open-chromatin domains features and signal distribution.

**ATAC-see signal quantification**. Fluorescence microscopy images were analyzed using an imageJ[104] macro as follow: first images were splitted by color. Then nuclei were segmented on DAPI max projection previous preprocessing and size-selection. ATTO590 signal was quantified in max projection images in nucleus-exclusive area after background subtraction. All experiments were median-normalized. Three independent experiments were performed. Number of nuclei quantified (*n*) per condition including the three replicates are provided in Supplementary Data 5.

**Motif-binding analysis with CentriMo**. Motif-binding analyses was performed with CentriMo (v5.0.2)[105] from MEME suite against all 579 binding motifs from JASPAR vertebrates[106] with default options. CentriMo perform Central Motif Enrichment Analysis (CMEA), which better identified binding motifs involved in the regulation of Chip-seq or ATAC-seq peaks. Position Weigh Matrixes (PWM) for SNAI1 and SMAD 5GC[107] binding motifs were also constructed and included in the analysis. For each category, enhancer sequences were shuffled five times for obtaining 5x random sequences with the same %CG content using scrambleFasta.pl from HOMER (v4.10.3) suite, and use that set of sequences as a background.

**Footprinting analysis**. To identify footprints in enhancer sequences wellington_footprints.py script from pyDNase suite (v0.2.4)[108] was used with -A option. Footprints were detected in TGFβ2h and TGFβ12h ATAC-seq data. Both ATAC-seq replicates for each condition were merged together for increasing the strength of footprint identification. For footprints enrichment analyses in different categories of chromatin accessibility first scrambleFasta.pl was used for obtaining 5x background sequences for each category. Then we look for motif-binding sites

in real footprints and background sequences with FIMO (v5.0.0)[109] using default option and $p$-value < 0.0001 as cutoff. Enrichment was calculated as:

$$\mathrm{Enr} = (\mathrm{footprints[motif]}/\mathrm{footprints[total]})/(\mathrm{footprints\_bg[motif]}/\mathrm{footprints\_bg[total]}) \quad (1)$$

where footprints[motif] is the number of footprints with a specific motif-binding sequence; footprints[total] is the number of total footprints; footprints_bg[motif] is the number of footprints with a specific motif-binding sequence in the background set of sequences; and footprints_bg[total] is the number of total footprints in the background set of sequences. $p$-values were calculated using fisher.test() from R stats package. Enrichment heatmap was plotted using gitools (v2.3.1) including most representatives motives.

Measurement of Tn5 integrations was calculated in Bioconductor using GenomicAlignments (v.1.20.1), Biostrings (v2.52.0) and soGGi (v.1.16.0) packages. Briefly, all ATAC-seq fragment smaller than 100 bp (nucleosome-free regions) were extracted from TGFβ2h BAM file. Then we resized our reads to 1 bp and make the shift of 4 or −5 bp depending on strand to adjust for expected shift from insertion of Tn5 transposase to produce cut-sites.

Footprinting analysis in regulated enhancers was performed as above but using as a category background all enhancers sequences that are not included in that category. TGFβ2h ATAC-seq data were used for footprints identification in early categories of regulated enhancers, and TGFβ12h for late categories.

**SMAD2/3 ChIP-seq analysis**. ChIP-seq data for SMAD2/3 in NMuMG cells were downloaded from GSE121254[36], and aligned against mm9 mouse reference genome using align() function from Rsubread package (v1.28.1) with type = 1, TH1 = 2 and unique = TRUE parameters. Peak calling was performed using macs2 callpeak using input as control file and default options. Then, peaks were filtered using −log10(FDR) ≥ 5 criteria and those peaks that overlap with blacklisted region of the genome were removed using bedtools subtract with -A options resulting in 8401 peaks. Peaks that overlap with the TSS ± 1000 bp of any transcript from mm9 UCSC KnownGene were discarded obtaining 7503 peaks, of which 5906 overlap with our enhancers. Overlapping analysis between SMAD2/3 bound enhancers and enhancers classified according to its increase in chromatin accessibility at TGFβ2h and/or AP-1 footprints were performed using vecsets (v1.2.1) package. $p$-values were calculated using dhyper() function from stats R package. ATAC-seq signal density graphs (Supplementary Fig. 3f) were plotted for TGFβ2h using computeMatrix reference-point from deeptools with -a 2000 -b 2000 options in different sets of regions: AP-1 footprinted enhancers, SMAD2/3 bound enhancers and both (enhancers with AP-1 footprints and SMAD2/3 binding).

**RNA-seq analysis**. Two independent biological replicates for each condition of RNA-seq data were primarily filtered using the FASTQ Toolkit v1.0.0 program. Data were aligned using subjunc function from Rsubread package, to map reads to the mm9 mouse reference genome using TH1 = 2 and unique=TRUE parameters. The downstream analysis was performed on bamfiles with duplicates removed using the samtools (v0.1.19) rmdup command. FeatureCounts() function from Rsubread package was used to assign reads to UCSC mm9 KnownGenes (miRNAs were discarded from the analysis) using GTF.featureType = "exon", GTF.attrType = "gene_id" and strandSpecific = 2 parameters on duplicate removed bamfiles. Then differential gene expression analysis was performed using the voom/limma (v.3.34.9) and edgeR (v.3.20.9) Bioconductor packages[110]. Genes that were expressed at ≥2 counts per million counted reads in ≥2 replicates were analyzed, resulting in 11081 genes out of 21240 UCSC KnownGene. CalcNormFactors() function using TMM method was used to normalize samples. Differentially expressed genes with $p$-value<0.05 (empirical BAYES moderated $t$-statistics test with adjustments for multiple comparison.) were classified according to:

Early-downregulated: log2FC(TGFβ2h_vs_veh) < −1 and not log2FC (TGFβ12h_vs_TGFβ2h) > 1

Early-upregulated: log2FC(TGFβ2h_vs_veh) > 1 and not log2FC (TGFβ12h_vs_TGFβ2h) < −1

Late-downregulated: log2FC(TGFβ12h_vs_veh) < −1 and not log2FC (TGFβ2h_vs_veh) < −1

Late-upregulated: log2FC(TGFβ12h_vs_veh) > 1 and not log2FC (TGFβ2h_vs_veh) > 1

Transient-downregulated: log2FC(TGFβ2h_vs_veh) < −1 and log2FC (TGFβ12h_vs_TGFβ2h) < −1

Transient-upregulated: log2FC(TGFβ2h_vs_veh) > 1 and log2FC (TGFβ12h_vs_TGFβ2h) > 1

TGFβ-independent: expressed genes that are not included in any of the categories above.

Because of low number of genes in transient categories, they were not taken into account for downstream analyses. For MA plots (Supplementary Fig. 4a), log2FC were plotted against average expression (log2CPM) for all samples for each comparison. For boxplots of Supplementary Fig. 4b, distribution of log2CPM of the two independent replicates were plotted. For expression heatmap of Fig. 2h log2FC were plotted for selected genes using gitools.

**Histone ChIP-seq analysis**. Two biological replicates of the ChIP-seq data for each condition and for each histone modification (H3K27ac, H3K4me1, and H3K4me3) were primarily filtered using the FASTQ Toolkit program. Data were aligned using align function from Rsubread package, to map reads to the mm9 mouse reference genome using type = 1, TH1 = 2 and unique = TRUE parameters. The downstream analysis was performed on bamfiles with duplicates removed using the samtools rmdup command. Information about read depth can be found in the reporting summary. Shushi package (v.1.16.0) was used for data visualization.

**Enhancer classification**. For creating the NMuMG cells enhancer repertoire first H3K27ac and H3K4me1 signal was measured by counting reads in full-length ATAC-seq peaks (29071 non-TSS peaks) using regionCounts() from csaw package. For H3K4me1, coordinates were increased ±500 bp because H3K4me1 showed a broad distribution. Then, CPM and RPKM were calculated using all counted reads in ATAC-seq peaks as library size and combining replicates. Peaks whose RPKM were larger than percentile 25 of all RPKM for H3K27ac or 10 for H4K4me1 were considered as having the mark. Enhancers without H3K4me1 nor H3K27ac in vehicle condition but that acquire these marks with TGFβ treatment were classified as latent. Enhancers with H3K4me1 but no H3K27ac were classified as primed and those with both marks, as active. To classify enhancers according to its regulation by TGFβ, we analyze H3K27ac dynamics in vehicle, 2 h and 12 h TGFβ-treated ChIP-seq data, considering both replicates independently and using a pipeline, including csaw and edgeR (v3.20.9) packages. Reads were counted in full-length ATAC-seq peaks as above and normalization factors were calculated as normOffset () function with option type = "loess" parameter. For differential binding analysis scaleOffset(), estimateDisp(), glmQLFit() and glmLRT() functions were used. Then enhancers were classified in early-activated, early-repressed, late-activated and late-repressed, following same criteria as differentially expressed genes. For each comparison we also consider a $p$-value < 0.05 and average log2CPM > 0. Because of low number of enhancers in transient categories, they weren't taking into account for downstream analyses.

EnrichedHeatmap package was used for drawing heatmaps. For Fig. 3a, rows were ordered according to decreasing H3K27ac CPM for TGFβ2h and all enhancers were plotted as summit ±500 bp increasing ±1500 bp from start and end. Boxplots of Supplementary Fig. 5, represents average CPM of the two replicates for each condition. Profiles screen-shot was constructed using sushi package (v1.16.0) on bedgraph constructed by bamCoverage from deeptools previous merging of replicates with samtools merge.

**ChromRNA-seq**. Two biological replicates for each condition of ChromRNA-seq data were primarily filtered using the FASTQ Toolkit program. Data were aligned using subjunc function from Rsubread package, to map reads to the mm9 mouse reference genome using TH1 = 2 and unique = TRUE parameters. The downstream analysis was performed on bamfiles with duplicates removed using the samtools rmdup command. As short size of eRNAs, ChromRNA-seq reads were counted in summit ±200 bp for intergenic enhancers using regionCounts(). From 29,071 enhancers, those which overlap with any UCSC KnownGene transcript bodies (5 kb extended from the Transcription termination site (TTS) to avoid termination read-through reads) were discarded, resulting in 12857 intergenic enhancers. After read counting, we calculated CPM of eRNA for each eRNA-expressing enhancer. Correlation between H3K27ac levels and eRNAs levels shown in Supplementary Fig. 6b was calculated using H3K27ac levels of the 12857 enhancers where eRNAs were counted. For correlation between increase in H3K27ac and increase in eRNA shown in Supplementary Fig. 6c, d, first we filtered by expression, obtaining 3636 eRNA-expressing intergenic enhancers. Then, differential expression analysis using csaw and edgeR was computed.

**Analysis of the location of TGFβ-regulated genes with respect to TGFβ-regulated enhancers**. To study the regulation of genes in the neighborhood of TGFβ-regulated enhancers, first we plotted log2FC distribution of genes adjacent to regulated enhancers. To determine position of genes with respect to enhancers, distances between gene TSSs and enhancers ATAC-seq peaks summit were calculated. Then genes were grouped according to: gene position with respect to the enhancers (upstream (-) or downstream(+)) and gene order with respect to the enhancer (1st to 4th). According to this classification, genes were called: enhancer-gene ± n, Eg ± n, with n = 1, 2, 3, and 4. Non-expressed genes were not taken into account for analysis. Randomization of enhancers was performed by shifting or by random sampling. For shifting randomization, lists of numbered enhancers for each recirculated chromosome were constructed and random enhancers were obtained by shifting enhancers 50 positions. So, if the regulated enhancer in at position n, the random enhancer is at position $n + 50$. For sampling randomization, all enhancers of each chromosome were randomly sampled. We obtained similar results with both randomization methods, however, we choose shifting method for considering it more conservative with genome structure. To study the dependency on enhancer-gene distance, we grouped genes located in 50 kb non-overlapping distance intervals (i.e., from summit to 50 kb, from 50 kb to 100 kb… upstream and downstream until 250 kb) from TGFβ-regulated enhancers of each enhancer category. The same was done for randomized enhancers. To study the correlation between change of activity of enhancers (measured as log2FC(H3K27ac)) and differential expression of the closest genes (log2FC), pairs of enhancer-closest

gene were ordered according to increasing log₂FC(H3K27ac) and grouped into 20 bins. Then, least squares fit analysis were performed to obtain r and slope. To study if enhancers operate in a continuous manner, first we subset those enhancers with one of their neighborhood genes (for example, Eg+2) robustly regulated (|log₂FC| > 1 and adjusted p-value < 0.05). Then, log₂FC distribution of the rest of genes in the neighborhood region (Eg-2, Eg-1, and Eg+1) were plotted. We only tested until Eg-2 and Eg+2 because of the small number of enhancers whose 3rd or 4th closest upstream or downstream gene is robustly regulated.

**Identification of clusters of co-regulated genes.** To identify clusters of co-regulated genes, first we sorted all genes according to 5′->3′ chromosomal order using gene TSS coordinates and we looked for robustly regulated pairs of genes from the same category (early-upregulated, early-downregulated, late-upregulated, or late-downregulated) that are strictly contiguous. To determine if the number of co-regulated pairs of genes was higher than expected by chance, we computed a random distribution of this score by randomly sampling a number of genes identical to the number of robustly regulated genes for each category and looking for pairs of contiguous genes. This process was iterated 5000 times, obtaining a distribution of random pairs of co-regulated genes. Probability of obtaining the real number of pairs of co-regulated genes given the random distributions was obtained using (https://keisan.casio.com/exec/system/1180573188).

**Analysis of gene co-regulation.** Similar to previous analysis, we numbered genes according to 5′->3′ chromosomal order using gene TSS coordinates (1, 2, …, n). Then, for every robustly TGFβ-regulated gene (|log₂FC| > 1 and adjusted p-value < 0.05), log₂FC of genes n ± k with k = 1, 2, 3, or 4, were plotted. Non-expressed genes were not taken into account for analysis. Random genes lists, per chromosome, were obtained by random sampling or by shifting 50 positions, as described above, with similar results. To study the effect of distance on co-regulation, we grouped genes into non-overlapping 50 kb distance intervals from robustly regulated gene TSSs (i.e., from TSS to 50 kb, from 50 kb to 100 kb… until 250 kb upstream and downstream) for each category and log₂FC distributions were plotted. Similar analysis was done for genes randomized by shifting or by sampling with similar results.

**Study of the influence of co-regulation in intergenic transcription.** To study the influence of co-regulation in intergenic transcription we determined the ChromRNA-seq signal in the neighborhood of TGFβ-regulated genes, previously all UCSC KnownGene transcript bodies (2 kb extended from the TSS and TTS to avoid termination read-through reads and divergent promoter transcription) were removed. Then, we counted reads of 250 kb (divided into 10 kb non-overlapping bins) upstream and downstream from a TGFβ robustly regulated gene TSS. For that, we used region.counts(). windowCounts() with bin = TRUE and width = 10,000 options. NormOffsets() was used to calculate normalization factors in libraries. Then we performed differential analysis bin to bin using estimateDisp() and glmQLFit() functions to obtain log₂FC. Similar analysis was performed using random genes (as described above) instead of TGFβ-regulated genes. Finally, log₂FC of random genes was subtracted from log₂FC of TGFβ-regulated genes, bin to bin, to obtain log₂FC over random values that were finally plotted.

**Study of the influence of co-regulation in histone modifications.** log₂FC of H3K27ac, H3K4me3, and H3K4me1 ChIP-seq signal in the neighborhood of TGFβ-regulated genes was performed as in the previous section. In this case 5 kb upstream and and downstream of the regulated gene TSS (or random gene) were removed, in order to eliminate promoter signal of the regulated gene. UCSC KnownGene transcript bodies coordinates were not removed from this analysis. Differential analysis was performed as in the previous section.

**Identification of TGFβ-regulatory domains.** To identify TGFβ-regulatory domains (TRD) we identified clusters of co-regulated genes, as above, using a different threshold of differentially expressed genes (|log₂FC| > 0.5 and p-value < 0.05) to include genes that are significantly regulated with TGFβ, although not robustly affected. Metaplots were created combining late and early TRDs for upregulation and downregulation and using computeMatrix scale-regions with -m 200,000 -bs 4000 -a 10,000 and -b 100,000 options. -bl option was used for calculating intergenic transcription (ChromRNA-seq) to avoid counting in all UCSC knownGene transcript bodies.

**Analysis of the relationship between TRDs and chromatin 3D organization.** We used TADs coordinates from NMuMG cells after 8 h of TGFβ treatment, published in ref. [49] after conversion to mm9 mouse gene assembly using UCSC liftover tool. Random TADs lists for each chromosome were obtained by recirculating the chromosome and shifting TAD borders by 10 Mb. To determine within TAD co-regulation, first TADs with less than three expressed genes were removed. Then, TADs were classified according to its percentage of expressed upregulated/downregulated genes (log₂FC > 0 or log₂FC < 0). For randomization, we entirely randomized positions of genes within TADs, conserving the same number of genes per TAD. Five hundred iterations were performed to obtain

random distributions, for statistical analysis. Observed/expected plot was created using the ratio real over the mean of the random distribution.

**Analysis of the role of TAD boundaries on gene co-regulation and enhancer-gene regulation.** To determine the expression correlation between two strictly contiguous genes (x, y), we sorted genes according to their 5′->3′ chromosomal order using gene TSSs and then we obtain all expression log₂FC pairs (log₂FC(x), log₂FC(y)) for all expressed genes, obtaining 6265 pairs. Then genes were sorted according to log₂FC(x) and grouped into 20 bins. For random plots we randomized position of all expressed genes and proceed in a similar way. To analyze the effect of TAD borders we extracted pairs (log₂FC(x), log₂FC(y)), where x and y genes were separated by a TAD border, obtaining 393 pairs. Then genes were sorted according to log₂FC(x) and grouped into 20 bins.

To study the effect of TAD borders in enhancer-gene regulation, enhancer-closest gene pairs separated by a TAD border were selected (1230 pairs). Then we extracted log₂FC(H3K27ac) of the enhancer and log₂FC(expression) of its closest gene that is separated by a TAD border, sorted by log₂FC(expression) and grouped into 20 bins. As random borders, we used the borders of random TADs that we describe in the previous section.

**Processing Hi-C data.** NMuMG TGFβ-treated for 8 h HiC data was obtained from GSE96033. Paired-end reads were aligned to the mm9 using bowtie2 (global parameters: –very-sensitive –L 30 –score-min L,−0.6,−0.2 –end-to-end–reorder; local parameters:–very-sensitive –L 20 –scoremin L,−0.6,−0.2 –end-to-end–reorder) through HiC-Pro software (v2.11.1)[111]. Unmapped reads, non-uniquely mapped reads and PCR duplicates were filtered and uniquely aligned reads were paired. Genome fragment distribution was obtained for DpnII restriction enzyme using as ligation sequence GATCGATC. Raw and ICE-normalized cis-contact matrices were assembled by binning paired reads into uniform 10-kb bins. HiTC Bioconductor package (v1.30.0) was used to analyze HiC interactions matrices. To determine intra-TRD interactions, interactions from raw and ICE-normalized matrices for TRD or random TRD coordinates were obtained using extractRegion() function. Random TRD coordinates were calculated inverting the start coordinate for each TRD and keeping the same length.

To analyze enhancer-promoter interactions first, we retrieve bins corresponding to TSS and enhancer midpoint coordinates. Then we obtained the interaction for each enhancer-promoter pair for raw matrix and grouped then according to: if the interaction occurs within the TRD (enhancer and promoter are located in the same TRD) or outside the TRD at different windows (enhancer is located within the TRD and genes are located at 0–100 kb, 100–250 kb, and 250–500 kb from the TRD upstream and downstream boundaries).

Directionality Index (DI) was calculated for each bin according to ref. [50]:

$$\text{DI} = \left( \frac{B - A}{|B - A|} \right) \cdot \left( \frac{(A - E)^2}{E} + \frac{(B - E)^2}{E} \right), \tag{2}$$

where A is the number of reads that map from a given 10 kb bin to the upstream 250 kb, B is the number of reads that map from the same 10 kb bin to the downstream 250 kb, and E, the expected number of reads under the null hypothesis, is equal to (A + B)/2. Then, directionality index metaplots were constructed for TADs and TRDs using proportional number of bin for each structure: 100 bins for TADs (as average size of TAD is 1 Mb) and 10 bins for TRDs. DI in boundaries was also plotted for TADs and TRDs using surrounding 100 bins (50 upstream and 50 downstream) and 30 bins respectively.

**CTCF metaplot analysis.** First, CTCF ChIP-seq data for mouse mammary gland tissue was downloaded from GSE74826 and was processed as described above for SMAD2/3 ChIP-seq data, finding 35360 CTCF peaks. Then we looked for motives in those peaks using fimo what allowed us to divide CTCF peaks according to its strand orientation: 17,800 in the forward and 17,850 in the reverse orientation. To identify CTCF footprints across the genome using our ATAC-seq data PIQ software[112] was used. CTCF density metaplots was constructed using ChIP-seq and footprints information in TADs and TRDs using proportional number of bin for each structure: 100 bins for TADs (as average size of TAD is 1 Mb) and 10 bins for TRDs. CTCF density in boundaries was also plotted for TADs and TRDs using surrounding 50 bins (25 upstream and 25 downstream).

**Statistical analysis.** Statistical and graphical data analyses were performed using either Prism 5 (Graphpad) software or R package. To determine the significance between two groups, comparisons were made using two-tailed Mann–Whitney non-parametric test by using wilcox.text() function form Stats R package. For correlation, the non-parametric Spearman coefficient was calculated and significance was calculated using Fisher exact test. Significance of enrichments were calculated with the Fisher exact test using fisher.test() function from R. Probabilities of overlapping were calculated using the hypergeometric distribution using dhyper() function from the stats R package. Other statistical methods are described above. The horizontal black line of the boxplot represents the median value, the box spans the 25th and 75th percentiles, and whiskers indicate 5th and 95th percentiles.

**Reporting summary**. Further information on research design is available in the Nature Research Reporting Summary linked to this article.

## Data availability

Sequencing data (raw data and processed files) that support the findings of this study have been deposited in NCBI GEO with the accession code "GSE140552". SMAD2/3 ChIP-seq data was downloaded as raw data from NCBI GEO accession code "GSE121254". HiC data was obtained from NCBI GEO accession code "GSE96033". CTCF ChIP-seq data was downloaded as a raw data from "GSE74826". All other relevant data supporting the key findings of this study are available within the article and its Supplementary Information files or from the corresponding author upon reasonable request. Source data are provided with this paper.

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

## Acknowledgements

We thank J. A. Pintor-Toro, F. Cortés-Ledesma, A. Lopez-Rivas, M. García-Domínguez, and M. Rodríguez-Paredes for reagents, critical reading of the manuscript, and continuous discussion. We thank P. Domínguez-Giménez from the CABIMER Microscopy Unit and E. Andújar and M. Pérez from the CABIMER Genomic Unit for technical assistance. This work was funded by the Spanish Ministry of Economy and Competitiveness (BFU2014-53543-P and BFU2017-85420-R to J.C. Reyes), the Junta de Andalucía (BIO-321), "Fundación Vencer El Cancer" (VEC) and the European Union (FEDER). CABIMER is a Center partially funded by the Junta de Andalucía. S.P was

supported by the FERO Foundation, La Caixa Foundation (LCF/PR/PR12/51070001) and Cellex Foundation.

## Author contributions

Conceptualization and methodology, J.A.G.-M. and J.C.R.; Investigation, J.A.G.-M. and M.C.-C.; Resources, F.K., S.P.; Writing, J.C.R. and J.A.G.-M.; Supervision, project administration, and funding acquisition: J.C.R.

## Competing interests

The authors declare no competing interests.
