## [Peer Review File · Nature Communications]

Reviewers' comments:

Reviewer #1 (Remarks to the Author):

I read with great interest the manuscript by Guerrero-Martínez et al. entitled "TGF β Promotes Widespread Enhancer Chromatin Opening and Operates on Genome Regulatory Domains". In this manuscript the authors investigate with high temporal resolution the chromatin and transcriptional changes that cells display shortly after TGF β treatment. Using this data the authors make two major and relevant observations: (i) most enhancers and promoters display a remarkable increase in chromatin accessibility regardless of whether they become activated or repressed by TGF β treatment; (ii) the chromatin and transcriptional changes at enhancer and target genes occur in regulatory domains (TGF β regulatory domains or TRD), distinct from TADs, in which single enhancers can coordinate the expression of several genes. Overall, the manuscript is interesting, novel, well-supported by the presented data and of broad relevance, thus making it a good candidate to be published in Nature Communications. My main concern is that, in its current format, the manuscript almost feels like two separate stories, which are both interesting but that are not follow up in sufficient depth from a mechanistic point of view. Therefore, as indicated below, I would suggest the authors to perform some additional experiments and analyses that, in my opinion, could help to generalize their observations and provide more mechanistic insights:

- I find the first part of the manuscript particularly interesting, relevant and intriguing. However, in order to more conclusively demonstrate that both activated and repressed enhancers display fast and pervasive chromatin opening and that the presented results are not due to some sort of technical artefact, I think the author should use some orthogonal transposase-independent method such as DNase-seq or FAIRE-seq.
- Is the pervasive chromatin the observed at enhancers transient?. The authors should also generate ATAC-seq data at a later point after TGF-beta treatment once EMT has been completed and the cells have acquired mesenchymal identity. It will be really interesting to test whether enhancers that become active remain accessible, whether for enhancers that become repressed, such opening could be truly transient.
- To increase the relevance of the manuscript, it would be interesting to evaluate whether the pervasive opening of enhancers is an exclusive feature of the response to TGF-beta or, alternatively, whether this could be a general response to other signalling pathways (e.g. BMP, WNT, etc). Similarly, it would be convenient if the authors use an alternative cell line/model to further show that the response to TGF-beta they observe is not exclusive to the cells they used.
- In Fig 2I the authors show Smad2/3 ChIP-seq data 1.5 hours after TGF-beta treatment. The data, in my opinion, is not very clearly presented. I think the authors should visualize the ChIP-seq data using heatmaps and/or boxplots for the different groups of enhancers. Moreover, it would be relevant to also generate Smad2/3 ChIP-seq data in untreated cells as well as at a later time point once EMT has been completed. This will reveal, as indicated below, whether Smad2/3 binding dynamics differs or not at activated (stable binding?) and repressed (transient binding?) enhancers. Moreover, for activated enhancers and given the potential cooperativity with AP-1, it is still possible that many of these enhancers are only transiently activated and are part of a rapid stress response that might not be truly relevant for EMT.
- Although interesting, I am not convinced that the concept of TRDs is completely novel, as it could basically correspond to sub-TAD chromatin conformations previously described and that have received various names in the literature: subTADs, insulated neighborhoods, 3D hubs, etc (Philips-Cremins et al, 2013; Downen et al, 2014; Miguel-Escalalada et al, 2019). In order to test this, the authors could use the available HiC data in their cell line and generate aggregate Hi-C plots around TRD to evaluate whether they observe any distinct topological feature. Similarly, they could use the Hi-C data to calculate insulation scores or directionality indexes around the TRD borders.
- Similarly, it would be highly relevant to evaluate whether the TRD borders are determined following similar or different rules with respect to previously described sub-TAD conformations. Namely, it would be relevant to generate CTCF ChIP-seq data in their cells and to evaluate

orientation of the underlying CTCF motifs to determine whether TRD borders are preferentially established by convergent CTCF motifs or, more interestingly, by some alternative mechanism. If the border coincide with CTCF sites, it would be interesting to delete some of them for at least one TRD and evaluate whether this spans the regulatory action of the TGF-beta responsive enhancers. - The authors indicate that the TGF-beta responsive enhancers display promiscuous activity. However, 3C-related methods have clearly shown over the last few years that the one gene-one enhancer rule is generally not true and any given enhancer can typically contact (and potentially regulate) multiple genes. Therefore, it would be relevant to perform 4C-seq experiments for one or two selected TRDs using a candidate enhancer as viewpoint to determine whether the enhancer establishes clear interactions with their target genes within the TRD vs those outside. Alternatively, the responsiveness of the genes within the TRD might not imply a direct/strong physical interaction with the enhancer but some alternative mechanism. Another possibility, instead of 4C-seq, is to use the available Hi-C data to try to solve this question.

Reviewer #2 (Remarks to the Author):

In this paper, Guerrero-Matinez et al integrated several genomic techniques to determine the mechanisms underlying how TGFb signaling controls chromatin accessibility and gene expression. There are two major claims from this study. First, TGFb treatment increased chromatin accessibility. These increased sites are associated with AP1 and Smad transcription factor binding. Second, many genes regulated in this process were found to be organized in "TGF-regulatory domains". The up- or down-regulation of the genes within each domain are influenced by TAD. Overall the paper is well-written, with a concise introduction and in-depth discussion. The statistical analyses included in this paper are strong. The techniques used in this paper were executed in high quality. However further evidence is needed in order to strengthen the conclusions.

For the first claim that "TGFb promotes widespread enhancer chromatin opening", all the experiments in this study were performed using the NMuMG cell line, a well-established model for EMT. In this model, the addition of TGFb in the presence of serum is sufficient to induce EMT-related gene expression within 6 hours (such as Xie et al., *Breast Cancer Res.* 2003; 5(6): R187–R198). In this paper, however, the NMuMG cells were serum-starved for 6 hours before TGFb treatment. It is understandable that this "serum starvation" strategy may allow specific dissection of TGFb's role in controlling chromatin accessibility. But most biological processes, such as EMT, occurs in the presence of other growth factors. Given the well-established EMT protocol for the NMuMG cell line, it is important to include the data of TGFb treatment without serum starvation, to show if TGFb addition can still induce increased chromatin accessibility. Both positive or negative results, from this suggested experiment, can provide valuable information to better frame the context of this claim.

Related to this first claim, it is striking that chromatin accessibility increased after only 10 minutes of TGFb treatment (Fig 1b). Furthermore, several AP1-family transcription factors were upregulated at 2 hours post-TGFb treatment at the mRNA level (Fig 1k). To better determine if the increased chromatin accessibility may be partially contributed by the AP1 factors, ATAC-seq data of 10 minutes postTGFb addition needs to be included to compare with the 2-hour time point. In addition, western blotting of AP1 factors need to be included to test if the protein levels, in addition to the mRNA levels, were increased at the 2-hour time point.

For the second claim related to "TGF-regulatory domains", it seems that only a fraction of the total genes regulated by TGFb are within these domains. What are the differences (such as biological functions, genomic locations, or chromatin features, etc) between the TGFb target genes inside versus outside these domains? In particular, are EMT genes located with or outside these TRDs? In addition, do all these TRDs contain Smad binding sites (based on the ChIP-seq data)?

A few additional minor comments:

1. Figure 1: Fig S2A is beautiful, and it should be moved to be part of Fig 1B. Fig 1G needs the number of ATAC-seq peaks labeled for each category. It is also intriguing that vehicle treatment alone increased ATAC-seq signal. Could the authors comment on that in the discussion?
2. Figure 4: Fig. S10A and S10B can be moved to the main Fig. 4, as Fig. 4A and Fig. 4B, to make a smoother transition.
3. Figure 5: Evidence demonstrating/validating the CRISPR KO efficiency of sgRNAs used in this figure needs to be included.

Reviewer #3 (Remarks to the Author):

Guerrero-Martinez et al. (Reyes) Nature Communications

The authors report on a genome-wide characterization of the genomic response to TGF- β using NMuMG cells, a model cell line for TGF- β -induced epithelial-mesenchymal transition. The genome-wide characterization of the enhancers reveals a substantial response, involving increased chromatin accessibility of a substantial portion of the enhancers throughout the genome, coregulation of genes adjacent to these enhancers, and key roles of AP1 (Jun-Fos) complexes in cooperation with Smad complexes.

Overall: This is an extensive analysis that combines Chip-Seq, RNA-Seq and bioinformatics approaches to characterize the genome-wide response, thus generating a database of genomic changes that occur in response to TGF- β . This information is likely to serve as a resource for further analyses by others.

While I am largely familiar with genomic and bioinformatics approaches, I am not expert in this area and therefore will not make a judgment as to whether better approaches or different parameters of analyses should have been used. I leave that to other reviewers. I am, however, very familiar with the biology of TGF- β , including the mechanisms of TGF- β signaling and Smad- (and non-Smad-) mediated transcription responses. It is from that perspective that I have concerns about the major conclusions reached by the authors based on their experimental design. Let me be specific:

The authors aim to evaluate the genomic response to TGF- β and present their study as if they indeed characterized the genomic response to TGF- β in NMuMG cells. Reading the manuscript and conclusions, it appears to me that the authors may not fully appreciate the fact that Smads act in cooperation with high-affinity DNA binding transcription factors that themselves are controlled by other signaling pathways (please consult recent reviews on this subject). In other words, the Smad-mediated genomic response, including the one presented here, fully depends on cooperation with such DNA binding transcription factors and other signaling pathways. As I was struck by the major contribution of AP1 complexes to the TGF- β /Smad-mediated response, and based on the original studies on Smad3/AP1 cooperation, I evaluated the experimental conditions used for TGF- β treatment, which unfortunately are not given with much detail. The way I read that first section of the Materials/Methods is that the cells were cultured in 10% serum with insulin, and were then serum-starved for 6 hrs. I assume therefore that the cells were maintained with insulin during the 6-hrs starvation, and that consequently the cells had activated insulin receptor signaling that leads to activation of AP1. Furthermore, a 6-hrs starvation is short for NMuMG cells, and I cannot be confident that indeed such short period is sufficient to silence the signaling in response to serum (growth factors). I also note that the authors did not present any evidence that the 6-hrs serum starvation led to downregulation of growth factor signaling through Erk MAPK or JNK, which act upstream from AP1. In fact, I assume that the cells had active growth factor signaling due to the presence of insulin. Consequently, it is highly likely that the authors did not evaluate the TGF- β response in its own right, but rather studied the response of TGF- β signaling in cooperation with insulin or growth factor signaling through Erk MAPK and JNK (and other pathways) that lead to

AP1 activation. Such response may or may not be relevant for contexts of Smad cooperation with other signaling pathways, but certainly should not be seen as "the" TGF- β response in NMuMG cells.

Considering this overall issue, the conclusion that TGF- β -induced non-Smad signaling is involved in the TGF- β -induced genomic response may not be valuable since the signaling inhibitors used are expected to act downstream from insulin and/or growth factors in serum, since 6 hr starvation of NMuMG cells may not have been sufficient to silence growth factor signaling, and in this way affect/inhibit AP1 complex formation and activation.

Finally, the authors have some minor grammatical inaccuracies in their writing, but this is a minor issue at this point.

First we want to thank Reviewers for their interesting and appropriated comments and suggestions. We think we have addressed most of the Reviewer's questions and the revised manuscript has improved very much in quality. Answers to the Reviewer's comments are in boldface. Figures for the Reviewers, designed Figures R1, R2...etc, are at the end of this .pdf document.

Reviewer #1 (Remarks to the Author):

I read with great interest the manuscript by Guerrero-Martínez et al. entitled "TGFβ Promotes Widespread Enhancer Chromatin Opening and Operates on Genome Regulatory Domains". In this manuscript the authors investigate with high temporal resolution the chromatin and transcriptional changes that cells display shortly after TGFbeta treatment. Using this data the authors make two major and relevant observations: (i) most enhancers and promoters display a remarkable increase in chromatin accessibility regardless of whether they become activated or repressed by TGFbeta treatment; (ii) the chromatin and transcriptional changes at enhancer and target genes occur in regulatory domains (TGFbeta regulatory domains or TRD), distinct from TADs, in which single enhancers can coordinate the expression of several genes. Overall, the manuscript is interesting, novel, well-supported by the presented data and of broad relevance, thus making it a good candidate to be published in Nature Communications. My main concern is that, in its current format, the manuscript almost feels like two separate stories, which are both interesting but that are not follow up in sufficient depth from a mechanistic point of view. Therefore, as indicated below, I would suggest the authors to perform some additional experiments and analyses that, in my opinion, could help to generalize their observations and provide more mechanistic insights:

- I find the first part of the manuscript particularly interesting, relevant and intriguing. However, in order to more conclusively demonstrate that both activated and repressed enhancers display fast and pervasive chromatin opening and that the presented results are not due to some sort of technical artefact, I think the author should use some orthogonal transposase-independent method such as DNase-seq or FAIRE-seq.

We understand the concern of the reviewer, so we have performed DNaseI-seq (vehicle and 2 h after TGFβ treatment) and compared the data with ATAC-seq data. After peak calling, we detected 23672 DNaseI hypersensitive peaks: 9978 of them at TSS and 13694 at non-TSS regions. 88.3% of the peaks (20908) were coincident with ATAC-peaks and 11.6% of the peaks (2764) were not associated with previously identified ATAC peaks (Figure R1a for the Reviewers). 76.8% of the DNaseI-ATAC common peaks increased their signal upon TGFβ treatment. Furthermore, a very good correlation was observed between DNaseI FC ($\log_2FC(TGF\beta 2h_vs_veh)$) and ATAC FC ($\log_2FC(TGF\beta 2h_vs_veh)$) (Figure

R1b). Only 58.6% of the DNaseI exclusive peaks increased upon TGF β treatment. However, these peaks were small peaks with a much lower number of reads (CPM) than the average of DNaseI-ATAC common peaks (Figures R1c and R1d). At this point we cannot say much about the molecular identity of these DNaseI exclusive peaks.

Therefore, a pervasive increase of DNaseI accessibility upon TGF β has been confirmed by a transposase-independent method. To further demonstrate this statement we have computed DNaseI-seq signal density around the enhancers ATAC-seq peaks shown in figure 2a of the new manuscript (previous figure 1G) divided into the four categories used in this plot ($FC < 1.5$, $1.5 \leq FC < 2$, $2 \leq FC < 4$ and $FC \geq 4$). As observed in Figure R1e, DNaseI-seq signal density increased in a similar proportion than the ATAC-seq signal, although the magnitude of the increment is lower.

The Reviewer was especially concerned about the chromatin opening in repressed enhancers (enhancers that lose H3K27Ac upon TGF β treatment). In Figure R1f we show three examples of enhancers that lose H3K27Ac upon TGF β treatment and show a strong increase in accessibility by ATAC-seq and by DNaseI-seq.

We have included in Supplementary figure 1c, d and e of the manuscript some of the panels shown in Figure R1, as a confirmation of ATAC-seq data. We have not described all the analysis neither have we included all the figures due to text words limitations.

- Is the pervasive chromatin the observed at enhancers transient? The authors should also generate ATAC-seq data at a later point after TGF-beta treatment once EMT has been completed and the cells have acquired mesenchymal identity. It will be really interesting to test whether enhancers that become active remain accessible, whether for enhancers that become repressed, such opening could be truly transient.

In order to address this interesting point raised by the Reviewer we have performed several experiments. The TGF β treatments described in the manuscript were performed under conditions of serum (and insulin) starvation in order to avoid signaling by other growth factors. Under these conditions NMuMG cells maintain normal viability and good shape for 2 or 12 h upon TGF β treatment. However, after 48 to 72 h cells start to get into apoptosis. Therefore, we could not investigate chromatin recovery under these conditions. As suggested by Reviewer #2, the EMT process can also be triggered by TGF β in the presence of serum in NMuMG cells and under these conditions most cells do not enter apoptosis, even after long periods of treatment. Therefore, experiments to investigate chromatin recovery after TGF β treatment were performed in complete medium (including serum and insulin). First, we verified by ATAC-seq that TGF β -dependent pervasive chromatin opening also occurs in complete medium (Figure R2a). We observed that under these circumstances the level of ATAC-seq signal in the non-treated cells (vehicle) is slightly higher than in experiments without serum (see also Figure R7 and Supplementary figure

2b of the revised manuscript), probably due to the presence of small amounts of dozens of growth factors (including TGF β) and hormones in the serum. However, we clearly observed a very significant increase of ATAC-seq signal 2 h after TGF β treatment. Interestingly, after 72 h we observed only a small decrease of ATAC-seq signal. Then, we hypothesized that most enhancers would remain opened while TGF β is present in the media. Therefore, we performed a series of experiments where TGF β -treatment was maintained during 2 hours, then washed out and then new complete medium but without additional TGF β was added. Under these conditions EMT advances normally, indicating that once the process is triggered, exogenous TGF β was not necessary anymore (Figure R2b). ATAC-seq demonstrated again only a minor chromatin recovery under these circumstances (2h of TGF β treatment + 72 h in the absence of exogenous TGF β) (Figure R2c). It is well known that TGF β pathway genes are under the positive control of TGF β ligands^{1,2}. In fact, our RNA-seq data demonstrated that *Tgfb1* and *Tgfbr1* genes are upregulated 2.3- and 5.5-fold, respectively (see Supplementary Table S1), indicating that NMuMG cells are able to produce endogenous TGF β that fuels the EMT process, once it is triggered exogenously. This may be the reason why signaling continues during the process and pervasive chromatin opening is maintained.

In order to demonstrate that the sustained increased accessibility is due to maintained autocrine signaling and that accessibility can be reversed once the signaling is suppressed, cells were first treated for 2 h with TGF β and then treated with the TGF β receptors inhibitor SB431542. Under these conditions we observed a complete reversion of the chromatin opening (Figure R2d).

After all these ATAC-seq preparatory experiments, we performed an ATAC-seq experiment as requested by the Reviewer, in order to investigate the behavior of the different categories of enhancers “at a later point after TGF-beta treatment”. We decided to stimulate the cells with TGF β for 2h and then to keep the cells in complete medium without TGF β for 72 h; time enough to have a completely mesenchymal phenotype. The data confirmed the results of the ATAC-seq experiments. After 72h most of the enhancers remains more accessible than in the non-treated cells (Figure R2e), and, in fact, most of them as accessible as at the 2h TGF β timepoint. Interestingly, the only enhancers that decreased slightly their level of ATAC-seq signal were those with the strongest increase of accessibility (FC \geq 4). Consistently, when the different enhancers categories (classified based on H3K27Ac fold change, as in Figure 3a of the manuscript) were analyzed, only early activated enhancers slightly decreased accessibility, respect to the 2h TGF β time point (Figure R2f). Figure R3 shows three examples of repressed enhancers that increase drastically accessibility after 2h of TGF β and remains accessible after additional 72 h in the absence of TGF β .

Taken together, these data indicate that, at least during the first 72 h after TGF β addition, the pervasive increase of chromatin accessibility is mostly maintained, even in repressed enhancers. Only early activated

enhancers lose some accessibility but stay at a higher level than in TGF β non-treated cells. However, complete inhibition of the TGF β pathway, aborts EMT and completely reverts chromatin opening.

We speculate that this behavior may be related to the reversible properties of EMT. It is possible that during EMT, repressed enhancers are not tightly closed through heterochromatin-like structures because enhancers have to be ready to be reactivated at any time for EMT reversion. It is well known that repressive and active marks in 'bivalent domains' co-occur at many promoters and at enhancers^{3,4}, in embryonic stem cells⁵. In fact, the EMT process generates cells with properties of stem cells⁶. A bivalent chromatin configuration at the ZEB1 promoter in cancer stem cells, modulated by TGF β , has been reported⁷. Therefore, it is possible that, in order to sustain plasticity during the EMT process, cells maintain most of their repressed enhancers in a bivalent state, not fully compacted, which may be the reason of the sustained accessible chromatin that we observe. However, we think that this hypothesis need to be properly investigated and it is preliminary to present the data in this paper in its present state. Therefore, we have decided do not include these experiments in the present manuscript. However, if the Editor and the Reviewer think that the inclusion of some of these data in the manuscript is essential we will do it.

- To increase the relevance of the manuscript, it would be interesting to evaluate whether the pervasive opening of enhancers is an exclusive feature of the response to TGF-beta or, alternatively, whether this could be a general response to other signalling pathways (e.g. BMP, WNT, etc). Similarly, it would be convenient if the authors use an alternative cell line/model to further show that the response to TGF-beta they observe is not exclusive to the cells they used.

To answer the question of the Reviewer we have chosen two epithelial cell lines of human origin, in this case. We performed ATAC-seq experiments in MCF7 (human breast cancer) and RPE1 (Human retinal pigment epithelial) cell lines. As described for NMuMG (Normal mouse mammary gland) in our manuscript, a generalized increase of ATAC-seq signal was observed upon TGF β treatment, although the kinetics of the process was slower with respect to the NMuMG cells (Supplementary Figure 2g of the Revised manuscript). It is important to mention that NMuMG cells carry out the fastest in vitro TGF β -dependent EMT described, at least to our knowledge, and, because of that, it is a commonly used model for this process^{8,9}. Thus, evident morphological EMT is observed after 16-18 h in NMuMG cells⁸, but changes start to be visible after less than 12h (Figure 1a of our manuscript). However, morphological changes in MCF7 or RPE1 cells take between 2 and 5 days⁹⁻¹¹. *SNAI1* is one of the master regulators of EMT process controlled by SMAD proteins. Maximum of *Snai1* mRNA induction occurs after 1h of TGF β treatment in NMuMG¹²; however, it takes 48 h in MCF7 cells¹³. In fact, defects in the phosphorylation of SMAD2 by TGF β in MCF7 cells have been reported¹⁴. These molecular data may contribute to understand the different kinetics observed in the

different cell lines.

We have included these results in the Supplementary Figure 2g.

The question about other signaling pathways raised by the Reviewer is very interesting. However, we think that it falls out of the scope of the present manuscript that it is dedicated to the characterization of TGF β -dependent enhancers.

- In Fig 2l the authors show Smad2/3 ChIP-seq data 1.5 hours after TGF-beta treatment. The data, in my opinion, is not very clearly presented. I think the authors should visualize the ChIP-seq data using heatmaps and/or boxplots for the different groups of enhancers. Moreover, it would be relevant to also generate Smad2/3 ChIP-seq data in untreated cells as well as at a later time point once EMT has been completed. This will reveal, as indicated below, whether Smad2/3 binding dynamics differs or not at activated (stable binding?) and repressed (transient binding?) enhancers. Moreover, for activated enhancers and given the potential cooperativity with AP-1, it is still possible that many of these enhancers are only transiently activated and are part of a rapid stress response that might not be truly relevant for EMT.

As recommended by the Reviewer we have included a heatmap, boxplots (Supplementary Figure 7 of the Revised version) and density plots of Smad2/3 ChIP-seq data for the different groups of enhancers (figure 3i of the Revised version).

The Reviewer comments that “it is still possible that many of these enhancers are only transiently activated”. The data presented above in this document suggest that chromatin opening is not transient, at least during the first 72 h after TGF β treatment. In the future, it would be worth studying different timepoints after weeks of EMT treatment. It has been shown that after a long period of time (2-3 weeks in the presence of TGF β) NMuMG cells suffer a process of transdifferentiation ¹⁵. In this situation cells maintain the mesenchymal phenotype and are able to proliferate. However, these long lasting treatments are not the objective of our manuscript. In fact, we wanted to concentrate in the early signaling upon TGF β treatment, and therefore, we find more appropriated to explore these interesting issues in a future work. Nevertheless, we agree with the reviewer that we cannot discard that pervasive enhancer opening and even activation of some of the enhancers is only part of a rapid stress response promoted by TGF β , but not relevant for EMT. We have included a sentence in this sense in the Discussion section.

We find interesting the purpose of making a study of the dynamic binding of Smad2/3 to the enhancers. However, we think that, given the results exposed above (lack of chromatin opening reversion after 72h), a correct analysis of the kinetics of this process would require several timepoints and a large amount of work and resources, so we will consider it as an important objective for the future.

- Although interesting, I am not convinced that the concept of TRDs is completely novel, as it could basically correspond to sub-TAD chromatin conformations previously described and that have received various names in the literature: subTADs, insulated neighborhoods, 3D hubs, etc (Philips-Cremins et al, 2013; Downen et al, 2014; Miguel-Escalalada et al, 2019). In order to test this, the authors could use the available HiC data in their cell line and generate aggregate Hi-C plots around TRD to evaluate whether they observe any distinct topological feature. Similarly, they could use the Hi-C data to calculate insulation scores or directionality indexes around the TRD borders.

TADs/subTADs and insulated neighborhoods are defined based on Hi-C and ChiA-PET data, respectively. However, TRD are defined based on co-regulation of two or more genes. Therefore, we think it would be misleading to use the same name as these previously described compartments. Nevertheless, it may be a correspondence between these structural compartments and the functional compartments. Therefore, as suggested by the Reviewer, we have used the available Hi-C data from the NMuMG cells to compute the average intra-TRD contacts and intra random-TRD contacts (See Figure R4a). TRD showed a slightly but non-significant higher number of intra-TRD ICE-normalized contacts per bin, than random regions of the same size. As indicated by the reviewer, we also calculated the aggregated directionality index of TRDs and TADs as control. As shown in Figure R4b, our analysis clearly identified a strong and sharp change of the directionality index at TADs borders. A much smaller and non-significant change of the directionality index was observed at TRD borders (Figure R4b). Therefore, we cannot clearly conclude that TRDs are subTAD structures. Directionality index analysis has been included as supplementary figure 15b and commented in the results section.

We agree with the Reviewer that the TRD concept is related to the insulated neighborhoods concept developed by Young. In fact, we mentioned this similarity in the Discussion section of the original and the revised version of the manuscript. However, insulated neighborhoods mostly contain one single gene together with its co-regulated enhancer¹⁶⁻¹⁹, although they may contain more than one gene. However, since TRDs are based on co-regulation of several genes, by definition, they contain at least two genes and often more than two. Furthermore, insulated neighborhoods are encompassed by CTCF sites, while we cannot clearly conclude that TRDs are surrounded by CTCF sites (see below). In addition, both structures (TRD and insulated neighborhoods) are defined based on completely different type of data. Therefore, we favor the idea of maintaining the TRD nomenclature in our paper.

- Similarly, it would be highly relevant to evaluate whether the TRD borders are determined following similar or different rules with respect to previously described sub-TAD conformations. Namely, it would be relevant to generate CTCF ChIP-seq data in their cells and to evaluate orientation of the underlying CTCF motifs to determine whether TRD borders are preferentially established

by convergent CTCF motifs or, more interestingly, by some alternative mechanism. If the border coincide with CTCF sites, it would be interesting to delete some of them for at least one TRD and evaluate whether this spans the regulatory action of the TGF-beta responsive enhancers.

We have used two strategies to evaluate the interesting question raised by the Reviewer. First, we performed TAD and TRD meta-analysis of CTCF occupancy using ChIP-seq data from mouse mammary gland tissue (GEO ID: GSE74826), corresponding to the origin of NMuMG cells. As expected, both TAD boundaries showed sharp peaks of CTCF density, where CTCF binding sites were identified in both orientations (Figure R5a). Although some increase of CTCF occupancy was observed at TRD boundaries, the magnitude of the increase was small and the plot was very noisy (Figure R5b). Statistical analysis of the data evidenced that differences were not significant respect to the neighborhoods. Second, we performed a similar analysis of occupancy using data of genomic footprinting analysis of our ATAC-seq data performed in NMuMG cells. In this case we could make the analysis in the three conditions used in our manuscript: vehicle, TGF β 2h and TGF β 12h. Results were similar in all the conditions, except that a higher general density was observed in the TGF β 12h data, probably due to the higher number of ATAC-seq reads in peaks of this condition which allows a better identification of footprints. In order to gain resolution, we also performed the analysis aggregating the three conditions. These data again evidenced clear peaks of CTCF at both TADs boundaries but no obviously higher density of CTCF at TRD borders (Figure R5c-f). Therefore, after this analysis, we cannot clearly conclude that TRDs are surrounded by CTCF sites. Part of this analysis has been included in the revised manuscript as Supplementary Figure 16.

- The authors indicate that the TGF-beta responsive enhancers display promiscuous activity. However, 3C-related methods have clearly shown over the last few years that the one gene-one enhancer rule is generally not true and any given enhancer can typically contact (and potentially regulate) multiple genes. Therefore, it would be relevant to perform 4C-seq experiments for one or two selected TRDs using a candidate enhancer as viewpoint to determine whether the enhancer establishes clear interactions with their target genes within the TRD vs those outside. Alternatively, the responsiveness of the genes within the TRD might not imply a direct/strong physical interaction with the enhancer but some alternative mechanism. Another possibility, instead of 4C-seq, is to use the available Hi-C data to try to solve this question.

In order to address the question raised by the Reviewer we have analyzed the available Hi-C data as suggested. While the number of reads of the available Hi-C experiment is not enough to get significant results for specific promoter-enhancer (E-P) pairs we have aggregated data from all E-P interactions inside and outside of the TRDs. Specifically, we have counted the number of chromatin contacts of every enhancer from all

TRDs with promoters inside the same TRD or at different distances from the TRD. The analysis clearly demonstrated that TRD enhancers mostly interact with TRD promoters of the same TRD. We have included this analysis as Supplementary figure 15a.

Reviewer #2 (Remarks to the Author):

In this paper, Guerrero-Martinez et al integrated several genomic techniques to determine the mechanisms underlying how TGF β signaling controls chromatin accessibility and gene expression. There are two major claims from this study. First, TGF β treatment increased chromatin accessibility. These increased sites are associated with AP1 and Smad transcription factor binding. Second, many genes regulated in this process were found to be organized in “TGF-regulatory domains”. The up- or down-regulation of the genes within each domain are influenced by TAD. Overall the paper is well-written, with a concise introduction and in-depth discussion. The statistical analyses included in this paper are strong. The techniques used in this paper were executed in high quality. However further evidence is needed in order to strengthen the conclusions.

For the first claim that “TGF β promotes widespread enhancer chromatin opening”, all the experiments in this study were performed using the NMuMG cell line, a well-established model for EMT. In this model, the addition of TGF β in the presence of serum is sufficient to induce EMT-related gene expression within 6 hours (such as Xie et al., Breast Cancer Res. 2003; 5(6): R187–R198). In this paper, however, the NMuMG cells were serum-starved for 6 hours before TGF β treatment. It is understandable that this “serum starvation” strategy may allow specific dissection of TGF β 's role in controlling chromatin accessibility. But most biological processes, such as EMT, occurs in the presence of other growth factors. Given the well-established EMT protocol for the NMuMG cell line, it is important to include the data of TGF β treatment without serum starvation, to show if TGF β addition can still induce increased chromatin accessibility. Both positive or negative results, from this suggested experiment, can provide valuable information to better frame the context of this claim.

We agree that it is important to know whether TGF β -dependent chromatin opening also occurs in cells that have not been subjected to serum starvation (including starvation of insulin, a supplement of the NMuMG culture medium). Therefore, we have performed in parallel, ATAC-seq experiments of cells serum-starved for 6 hours or non-starved. As shown in the new Supplementary Figure 2b, results were very similar in serum starved and in non-starved cells. Kinetics was also similar, since TGF β effect was already observed 10 min after addition of the growth factor and maintained after 2h. Vehicle non-starved cells presented a small but significant higher signal than vehicle starved cells, consistent with the fact that serum contains dozens of growth factors (including TGF β) at small concentrations, that may be involved in opening hundreds of enhancers and promoters. In addition, as commented in the answer to question 2 of Reviewer #1, all ATAC-seq experiments shown in Figure R2

were performed in non-starved cells.

Furthermore, we have performed an ATAC-seq experiment in non-starved cells (already described in the answer to question 2 of Reviewer #1 and Figure R2). We have compared the ATAC-seq data performed after 6 h of starvation with the experiment performed in non-starved cells at the 2h after TGF β timepoint. We found that 73% of the non-TSS ATAC-seq peaks in non-starved cells were identical to those of starved cells (new Supplementary Figure 2c). Importantly, 91.38% and 97.96% of the peaks from non-starved and starved cells, respectively, increased accessibility 2 h after TGF β addition. In fact, we observed a very good correlation between accessibility FCs (TGF β 2h_vs_veh) between both conditions (new Supplementary Figure 2d). Furthermore, new Supplementary Figure 2e shows that non-starvation specific ATAC peaks (those that were not present in the ATAC-seq experiment performed under 6h of starvation) also increased accessibility upon TGF β . This figure also shows that, in average, chromatin accessibility increase was slightly higher in the serum starvation experiment than in the experiment performed in normal serum. Therefore, these data demonstrate that under normal serum conditions TGF β also promotes a massive increase of chromatin accessibility, as we describe in the manuscript for 6 h serum-starved cells. We include this conclusion in the Revised manuscript.

Furthermore, Reviewer #3 suggested that we should have used a longer serum starvation for our experiments. We performed ATAC-seq experiments after 24 and 48 h of serum starvation followed by 2h of TGF β treatment (Supplementary figure 2f). The results were also similar, demonstrating that although serum may have a quantitative role in some specific enhancers, the general conclusion that TGF β promotes a pervasive increase of accessibility is maintained under all the conditions tested.

Related to this first claim, it is striking that chromatin accessibility increased after only 10 minutes of TGF β treatment (Fig 1b). Furthermore, several AP1-family transcription factors were upregulated at 2 hours post-TGF β treatment at the mRNA level (Fig 1k). To better determine if the increased chromatin accessibility may be partially contributed by the AP1 factors, ATAC-seq data of 10 minutes post-TGF β addition needs to be included to compare with the 2-hour time point. In addition, western blotting of AP1 factors need to be included to test if the protein levels, in addition to the mRNA levels, were increased at the 2-hour time point.

As requested by the Reviewer, we have performed new ATAC-seq experiments after 10 minutes of TGF β or vehicle treatment. At this timepoint we also observed a massive chromatin opening (See heatmaps of the new Figure 2b). Interestingly, enhancers with strong accessibility increase at 10 min (FC > 4) were enriched in footprints with motifs of the ETS family including ELK4, ELK1, ETS1 transcription factors, all of them

under the control of the ERK pathway (See heatmap of the new 10 minutes panel of Figure 2c). In fact, AP-1 factors are less enriched in this category than ETS factors supporting that at 10 minutes there is an effect of ERK1 through ETS factors and independently of AP-1. This result is consistent with the result of the western blotting of AP-1 factors proposed by the Reviewer. Thus, we have performed western blotting of Jun and Fos after 10 minutes and 2h. As shown in the new Figure 2g, both AP-1 factors increased upon 2h of TGF β treatment, but not at the 10 minutes timepoint. Probably, for some members of the AP-1 family, significant levels are already present before TGF β addition (as observed in the case of Jun in our western), but others are present at very low levels before induction. Shortly after induction, but later than 10 min, levels of most AP1 factors increase, which may explain the high enrichment of AP1 footprints at the 2h timepoint. We have included these results and discussed the possibility of a role in chromatin opening of the ERK-ETS pathway at the 10 min timepoint, at least in some enhancers.

For the second claim related to “TGF-regulatory domains”, it seems that only a fraction of the total genes regulated by TGF β are within these domains. What are the differences (such as biological functions, genomic locations, or chromatin features, etc) between the TGF β target genes inside versus outside these domains? In particular, are EMT genes located with or outside these TRDs? In addition, do all these TRDs contain Smad binding sites (based on the ChIP-seq data)?

As suggested by the Reviewer we have analyzed several aspects of TGF β -regulated TRD genes and non-TRD genes. First we analyzed Gene Ontology of both set of genes. In both cases we found categories related to TGF β and EMT, such as “cell cycle”, “apoptotic process”, and TGF β signaling pathway categories (Supplementary Figure 12a). Some important TGF β signaling or EMT-related genes such as *Tgfb1*, *Dab2*, *Epb4115*, or *Id3* are in TRDs and other such as *Snai1*, *Smad7*, *Fn1* or *Cdh1* are out of TRDs. Next, we analyzed the chromosomal positions of both sets of genes and we did not find any special distribution or association with specific chromosomes (data not shown). However, we found that TRD genes are in regions with higher gene density than non-TRD genes. As a consequence, the gene-gene distances of TRD genes (calculated as the distance between one gene and its closest gene) were shorter than those of non-TRD genes (Supplementary Figure 12b). The distance between a gene and the closest co-regulated enhancer was also shorter in TRD than in non-TRD genes (Supplementary Figure 12c). However, there were no differences when distances between a gene and the closest anti-regulated enhancer were computed (Supplementary Figure 12d). These data suggest that TRDs are compact domains. We have included all these new Supplementary figures in the manuscript.

Next, we computed mRNA levels and TGF β -dependent fold change of TRD and non-TRD genes (Supplementary Figure 12e,f). Total mRNA levels, counted as RNA-seq CPMs, were similar in TRD and non-TRD

genes, with a slightly lower average expression of TRD genes after 12 h of TGF β . TGF β -dependent expression changes were slightly more robust in TRD genes than in non-TRD genes. Thus, TRD upregulated genes were, in average, slightly more upregulated by TGF β than non-TRD genes. Equally, TRD repressed genes were, in average, more downregulated than non-TRD genes. Despite the observation of this tendency in all the categories, it was only statistically significant in the case of downregulated genes. These results have been included as Supplementary Figure 12e, f. Similar Chromatin accessibility (Figure R6a, b), H3K27Ac (Figure R6c-g), H3K4me1 (not shown) and H3K4me3 (not shown) between TRD and non-TRD enhancers were observed. These negative results were not included as supplementary Figures, since they do not provide interesting information, from our point of view, and also due to text space limitations.

A few additional minor comments:

1. Figure 1: Fig S2A is beautiful, and it should be moved to be part of Fig 1B. Fig 1G needs the number of ATAC-seq peaks labeled for each category. It is also intriguing that vehicle treatment alone increased ATAC-seq signal. Could the authors comment on that in the discussion?

Fig. S2A of the original manuscript has been now moved to Figure 1c, as suggested by the Reviewer, together with the plots of characterization of the confocal microscopy signal. Number of peaks has been included in the former Figure 1g, that it is now Figure 2a.

Vehicle treatment does not increase ATAC-seq signal. As expected, vehicle-treated cells have a basal level of chromatin accessibility. In fact, panel VEH of the former Fig. 1g, (now Figure 2a) corresponds to the basal ATAC-seq accessibility level. In ATAC-seq experiments it is also visible a basal level of accessibility in vehicle. This should not be confused with the “Neg control” line in ATAC-seq experiments, where the ATAC reaction is performed in the presence of EDTA, which inhibits transposition.

2. Figure 4: Fig. S10A and S10B can be moved to the main Fig. 4, as Fig. 4A and Fig. 4B, to make a smoother transition.

Fig. S10A and S10B of the original version has been moved to the new figure 5 (which corresponds to the original Figure 4).

3. Figure 5: Evidence demonstrating/validating the CRISPR KO efficiency of sgRNAs used in this figure needs to be included.

First, we wanted to clarify that we have not used CRISPR KO, but CRISPRi (CRISPR interference), which allows the functional epigenetic inactivation of the region where the dCas9-KRAB fusion protein is targeted by the sgRNA²⁰. In order to demonstrate sgRNAs efficiency we have performed two types of experiments: First, we performed CHIP-PCR of dCas9-KRAB at the six different enhancers, using anti-Cas9 antibodies. CHIP

experiments clearly demonstrated targeting of the sgRNA-dCas9-KRAB complex to the expected enhancer regions. Second, we show that targeting to dCas9-KRAB promotes a decrease of H3K27Ac, confirming the expected repressive activity of the system. These important controls have been included in the new Supplementary figure 13.

Reviewer #3 (Remarks to the Author):

Guerrero-Martinez et al. (Reyes) Nature Communications

The authors report on a genome-wide characterization of the genomic response to TGF- β using NMuMG cells, a model cell line for TGF- β -induced epithelial-mesenchymal transition. The genome-wide characterization of the enhancers reveals a substantial response, involving increased chromatin accessibility of a substantial portion of the enhancers throughout the genome, coregulation of genes adjacent to these enhancers, and key roles of AP1 (Jun-Fos) complexes in cooperation with Smad complexes.

Overall: This is an extensive analysis that combines Chip-Seq, RNA-Seq and bioinformatics approaches to characterize the genome-wide response, thus generating a database of genomic changes that occur in response to TGF- β . This information is likely to serve as a resource for further analyses by others. While I am largely familiar with genomic and bioinformatics approaches, I am not expert in this area and therefore will not make a judgment as to whether better approaches or different parameters of analyses should have been used. I leave that to other reviewers. I am, however, very familiar with the biology of TGF- β , including the mechanisms of TGF- β signaling and Smad- (and non-Smad-) mediated transcription responses. It is from that perspective that I have concerns about the major conclusions reached by the authors based on their experimental design. Let me be specific:

The authors aim to evaluate the genomic response to TGF- β and present their study as if they indeed characterized the genomic response to TGF- β in NMuMG cells. Reading the manuscript and conclusions, it appears to me that the authors may not fully appreciate the fact that Smads act in cooperation with high-affinity DNA binding transcription factors that themselves are controlled by other signaling pathways (please consult recent reviews on this subject). In other words, the Smad-mediated genomic response, including the one presented here, fully depends on cooperation with such DNA binding transcription factors and other signaling pathways. As I was struck by the major contribution of AP1 complexes to the TGF- β /Smad-mediated response, and based on the original studies on Smad3/AP1 cooperation, I evaluated the experimental conditions used for TGF- β treatment, which unfortunately are not given with much detail. The way I read that first section of the Materials/Methods is that the cells were cultured in 10% serum with insulin, and were then serum-starved for 6 hrs. I assume therefore that the cells were maintained with insulin during the

6-hrs starvation, and that consequently the cells had activated insulin receptor signaling that leads to activation of AP1.

We apologize for not mentioning that insulin was also removed during the 6 h of serum starvation in Materials and Methods. We have now corrected the mistake. So, TGF β -treatments presented in the original manuscript were all performed after a period of 6 h without insulin and without serum. Nevertheless, we have performed a new ATAC-seq experiment to investigate the effect of insulin in the pervasive TGF β -dependent chromatin opening that we describe. As shown in Figure R7, presence of insulin during the 6 h of starvation has no effect in chromatin opening assayed by ATAC-seq.

Furthermore, a 6-hrs starvation is short for NMuMG cells, and I cannot be confident that indeed such short period is sufficient to silence the signaling in response to serum (growth factors). I also note that the authors did not present any evidence that the 6-hrs serum starvation led to downregulation of growth factor signaling through Erk MAPK or JNK, which act upstream from AP1.

We did two experiments to try to address this concern of the Reviewer. First, we performed western blotting using antibodies against Erk and phosphor-Erk. Phosphor-Erk signal significantly decreased after 6 h of serum (and insulin) starvation. We also verified that the level of activated Erk increased drastically 10 minutes after TGF β treatment, but this increase was transitory and after 2h of treatment with TGF β , levels of phosphor-Erk decreased to the same level of normal proliferating cells. We believe that this is a very appropriated control for our manuscript, and we have included this western as Supplementary Figure 2a.

Nevertheless, we understand the concern of the reviewer about the starvation period. Because of that, we performed an ATAC-seq experiment to investigate whether a longer serum starvation has an effect on the pervasive increase of chromatin accessibility provoked by TGF β . As shown in the new Supplementary figure 2f, cells starved for 24 or 48 h (serum and insulin starvation) still respond promoting a strong increase of chromatin accessibility upon TGF β treatment.

In fact, I assume that the cells had active growth factor signaling due to the presence of insulin. Consequently, it is highly likely that the authors did not evaluate the TGF- β response in its own right, but rather studied the response of TGF- β signaling in cooperation with insulin or growth factor signaling through Erk MAPK and JNK (and other pathways) that lead to AP1 activation. Such response may or may not be relevant for contexts of Smad cooperation with other signaling pathways, but certainly should not be seen as “the” TGF- β response in NMuMG cells.

Considering this overall issue, the conclusion that TGF- β -induced non-Smad signaling is involved in the TGF- β -induced genomic response may not be valuable since the signaling inhibitors used are expected to act downstream from insulin and/or growth factors in serum, since 6 hr starvation of NMuMG

cells may not have been sufficient to silence growth factor signaling, and in this way affect/inhibit AP1 complex formation and activation.

In the revised version of the manuscript, we have now clarified that all TGF β experiments (except when indicated) were performed after 6 hours of serum and insulin starvation. Furthermore, we show that the presence or absence of insulin during the 6 h of serum starvation does not change the accessibility effect of TGF β . We also show that this starvation period is able to decrease Erk activity. We confirm that the increase of accessibility by TGF β is not affected by a longer growth factors starvation period (24h or 48h). Interestingly Reviewer 2 suggested performing the experiments in complete medium (without serum starvation). As can be observed in the new supplementary figures 2b-e, we have also shown, by ATAC-seq and ATAC-seq, that TGF β -dependent increase of accessibility is very similar in the presence of normal serum. Therefore, we think that we have demonstrated that the strong effect promoted by TGF β in chromatin accessibility is independent of the presence of serum (and insulin).

Finally, the authors have some minor grammatical inaccuracies in their writing, but this is a minor issue at this point.

Grammatical inaccuracies have been revised by a English editor.

References of the Answer to the Reviewers.

1. Scheel, C. *et al.* Paracrine and autocrine signals induce and maintain mesenchymal and stem cell states in the breast. *Cell* **145**, 926-940 (2011).
2. Oft, M. *et al.* TGF-beta1 and Ha-Ras collaborate in modulating the phenotypic plasticity and invasiveness of epithelial tumor cells. *Genes Dev* **10**, 2462-2477 (1996).
3. Charlet, J. *et al.* Bivalent Regions of Cytosine Methylation and H3K27 Acetylation Suggest an Active Role for DNA Methylation at Enhancers. *Mol Cell* **62**, 422-431 (2016).
4. Bernhart, S. H. *et al.* Changes of bivalent chromatin coincide with increased expression of developmental genes in cancer. *Sci Rep* **6**, 37393 (2016).
5. Bernstein, B. E. *et al.* A bivalent chromatin structure marks key developmental genes in embryonic stem cells. *Cell* **125**, 315-326 (2006).
6. Mani, S. A. *et al.* The epithelial-mesenchymal transition generates cells with properties of stem cells. *Cell* **133**, 704-715 (2008).
7. Chaffer, C. L. *et al.* Poised chromatin at the ZEB1 promoter enables breast cancer cell plasticity and enhances tumorigenicity. *Cell* **154**, 61-74 (2013).
8. Miettinen, P. J., Ebner, R., Lopez, A. R. & Derynck, R. TGF-beta induced transdifferentiation of mammary epithelial cells to mesenchymal cells: involvement of type I receptors. *J Cell Biol* **127**, 2021-2036 (1994).

9. Moreno-Bueno, G. *et al.* The morphological and molecular features of the epithelial-to-mesenchymal transition. *Nature protocols* **4**, 1591-1613 (2009).
10. Gamulescu, M. A. *et al.* Transforming growth factor beta2-induced myofibroblastic differentiation of human retinal pigment epithelial cells: regulation by extracellular matrix proteins and hepatocyte growth factor. *Experimental eye research* **83**, 212-222 (2006).
11. Tamiya, S., Liu, L. & Kaplan, H. J. Epithelial-mesenchymal transition and proliferation of retinal pigment epithelial cells initiated upon loss of cell-cell contact. *Invest Ophthalmol Vis Sci* **51**, 2755-2763 (2010).
12. Dave, N. *et al.* Functional cooperation between Snail1 and twist in the regulation of ZEB1 expression during epithelial to mesenchymal transition. *J Biol Chem* **286**, 12024-12032 (2011).
13. Tian, M. & Schiemann, W. P. TGF-beta Stimulation of EMT Programs Elicits Non-genomic ER-alpha Activity and Anti-estrogen Resistance in Breast Cancer Cells. *J Cancer Metastasis Treat* **3**, 150-160 (2017).
14. Brown, K. A. *et al.* Induction by transforming growth factor-beta1 of epithelial to mesenchymal transition is a rare event in vitro. *Breast Cancer Res* **6**, R215-231 (2004).
15. Gal, A. *et al.* Sustained TGF beta exposure suppresses Smad and non-Smad signalling in mammary epithelial cells, leading to EMT and inhibition of growth arrest and apoptosis. *Oncogene* **27**, 1218-1230 (2008).
16. Downen, J. M. *et al.* Control of cell identity genes occurs in insulated neighborhoods in mammalian chromosomes. *Cell* **159**, 374-387 (2014).
17. Hnisz, D., Day, D. S. & Young, R. A. Insulated Neighborhoods: Structural and Functional Units of Mammalian Gene Control. *Cell* **167**, 1188-1200 (2016).
18. Ji, X. *et al.* 3D Chromosome Regulatory Landscape of Human Pluripotent Cells. *Cell stem cell* **18**, 262-275 (2016).
19. Sun, F. *et al.* Promoter-Enhancer Communication Occurs Primarily within Insulated Neighborhoods. *Mol Cell* **73**, 250-263 e255 (2019).
20. Gilbert, L. A. *et al.* CRISPR-mediated modular RNA-guided regulation of transcription in eukaryotes. *Cell* **154**, 442-451 (2013).

Figure R1

Figure R1. **a** Overlapping between DNaseI-seq and ATAC-seq peaks. **b** Correlation between changes of ATAC-seq and DNaseI-seq signals upon TGF β addition ($\log_2\text{FC}(\text{TGF}\beta 2\text{h_vs_veh})$), of the 20908 common regions of open chromatin found using both techniques. **c** MA plot of DNaseI-seq signal (CPM) of DNaseI-ATAC common (blue) and DNaseI exclusive (green) peaks versus DNaseI-seq fold changes ($\log_2\text{FC}(\text{TGF}\beta 2\text{h_vs_veh})$). **d** Left panel. Boxplot of DNaseI-seq fold changes ($\log_2\text{FC}(\text{TGF}\beta 2\text{h_vs_veh})$) of DNaseI-ATAC common and DNaseI exclusive peaks. Right panel. Boxplot of DNaseI-seq signal (CPM) of DNaseI-ATAC common and DNaseI exclusive peaks. *** $p \leq 0.001$ (Mann-Whitney non-parametric test). **e** DNaseI-seq signal density plots of vehicle and 2h TGF β at non-TSS ATAC-seq peaks. The four curves correspond to the ATAC-peaks categories defined in Figure 2a of the manuscript. **f** Three examples of enhancers repressed by TGF β (decrease of H3K27Ac at the TGF β 2h time point) that gain accessibility assayed by ATAC-seq and by DNaseI-seq.

Figure R2

Figure R2. For all the TGF β treatments described in this figure, NMuMG cells were not serum starved previous to TGF β addition. **a** Quantification of ATAC-seq under the indicated conditions. Cells were exposed during the indicated time to TGF β . Negative control and vehicle as indicated in Figure 1 of the manuscript. **b** Cells perform EMT after only 2 h of TGF β treatment, even if TGF β is removed during the following 72 h. **c** Quantification of ATAC-seq under the indicated conditions. Cells were exposed during 2h to TGF β , then washed with PBS and then cultured for 72 additional hours in complete medium without exogenously added TGF β . **d** Quantification of ATAC-seq under the indicated conditions. Cells were exposed during 2h to TGF β and then treated with 10 μ M SB431542 (SB) for 2 additional hours. Vehicle treated cells were also treated with SB431542 as control. **e** ATAC-seq experiments were performed as follow: serum non starved cells were treated during 2h with TGF β or vehicle, then washed with PBS and then cultured for 72 additional hours in complete medium without exogenously added TGF β . Heatmaps are shown. **f** Boxplots showing quantification of ATAC-seq signal in the different categories of enhancers shown in figure 3a of the manuscript. TGF β 2h, TGF β treatment for 2h; TGF β REC, TGF β treatment for 2h plus RECOVERY of TGF β during 72 additional h in complete medium without exogenously added TGF β .

Figure R3

Figure R3. Three examples of enhancers repressed by TGFβ with increased accessibility at 2h after TGFβ and that maintains highly accessible chromatin after 72 h of culture in complete medium without exogenously added TGFβ.

Figure R4

a

Hi-C interactions intra y extra TRD

b

Directionality index

Figure R4. Hi-C data from NMuMG (GEO: [GSM3243024](https://www.ncbi.nlm.nih.gov/geo/query/acc.cgi?acc=GSM3243024)) were analyzed at 10 kb resolution. **a** Boxplot showing quantification of Hi-C interactions intra-TRD or intra-random TRD, using row data or ICE normalized data. **b** Meta-TAD and meta-TRD directionality index analysis. Lower panel: Directionality index density was calculated as indicated in Methods of the manuscript. TADs were divided into 100 bins and TRD were divided into 10 bins, proportional to the about 1 Mb and 100 kb average size of each structure. Upper panel: Significance (p-value of Mann–Whitney Test) of the difference between two consecutive bins. Red line indicates $p = 0.05$.

Figure R5

CTCF ChIP-seq

CTCF footprint analysis at NMuMG

Figure R5. CTCF distribution at TAD and TRD borders. Meta-TAD (**a, c, e**) and meta-TRD (**b, d, f**) CTCF density analysis. Density of CTCF binding sites in the plus and in the minus strand is represented. **a, b** CTCF occupancy was determined by using ChIP-seq data from mouse mammary gland tissue (GEO ID: GSE74826). CTCF motif orientation was determined through CTCF sites identification at the CTCF peaks, by using HOMER. **c-f** CTCF occupancy and motif orientation was determined using our ATAC-seq footprinting analysis. **c, d** ATAC-seq reads from vehicle, 2h TGF β and 12h TGF β data were analyzed separately. **e, f** ATAC-seq reads from the different experimental conditions were pooled in order to have better resolution.

Figure R6. Levels of ATAC-seq signal and H3K27Ac of enhancers located in TRD or out of TRDs.

Figure R7

Figure R7. Effect of presence or not of insulin on TGFβ–dependent increase of chromatin accessibility. Quantification of ATAC-seq under the indicated conditions. Cells were subjected to a period of 6 h in the following conditions prior to TGFβ treatment for the indicated time: -FBS/-Ins, serum and insulin starvation; -FBS/+Ins, serum starvation but normal levels of insulin; +FBS/+ins, complete medium (DMEM medium supplemented with serum and insulin). *** $p \leq 0.001$ (Mann-Whitney non-parametric test)

REVIEWERS' COMMENTS

Reviewer #1 (Remarks to the Author):

I would like to thank the authors for a thorough and constructive revision. They have properly addressed most of my concerns and, thus, I now recommend publication in Nature Communications. I would suggest to include in the manuscript the new results regarding chromatin accessibility at longer times following EMT.

Reviewer #2 (Remarks to the Author):

The authors did an outstanding job addressing all my points. I don't have any additional questions. Great work!

Reviewer #3 (Remarks to the Author):

Guerrero-Martinez et al. (Reyes) Nature Communications Revised

The authors report on a genome-wide characterization of the genomic response to TGF- β using NMuMG cells, a model cell line for TGF- β -induced epithelial-mesenchymal transition. The genome-wide characterization of the enhancers reveals a substantial response, involving increased chromatin accessibility of a substantial portion of the enhancers throughout the genome, coregulation of genes adjacent to these enhancers, and key roles of AP1 (Jun-Fos) complexes in cooperation with Smad complexes.

Overall: This is an extensive analysis that combines Chip-Seq, RNA-Seq and bioinformatics approaches to characterize the genome-wide response, thus generating a database of genomic changes that occur in response to TGF- β . This information is likely to serve as a resource for further analyses by others. I am very familiar with the biology of TGF- β , including mechanisms of TGF- β signaling and Smad- (and non-Smad-) mediated transcription responses, and reviewed the revised manuscript from that perspective. The authors have satisfactorily addressed my comments, both in the rebuttal and in the revised manuscript.

Some minor comments remain:

- The term TAD known to most in the fields of genomic analyses has been extensively used, but has not been explained (not even what the abbreviation stands for).
- In the Introduction (line 76), the TGF- β receptors are mentioned as serine-threonine kinases; however, they are dual specificity kinases as has been well established starting 26-27 years ago. I am obviously aware that many (some of them stubbornly) do not seem to realize this and propose them to be Ser/Thr kinases, because they phosphorylate R-Smads on Ser.
- line 445 "stablishes" should be "establishes".
- From time to time the authors talk about Smad2/3 occupation of enhancers. They obviously do so because the antibody used recognizes Smad2 and Smad3. But let's be realistic, most TGF- β responses (with very few exceptions) are Smad3-mediated. I realize that one has to be scientifically correct but maybe it is worth explaining to the reader that Smad3 is the major TGF- β signaling effector, so that the non-aficionado reader does not have misconceptions.
- ref 35 is incomplete.

First we want to thank Reviewers for their suggestions that have contributed to the final acceptance of this manuscript. Answers to the Reviewer's comments are in boldface.

Reviewer #1 (Remarks to the Author):

I would like to thank the authors for a thorough and constructive revision. They have properly addressed most of my concerns and, thus, I now recommend publication in Nature Communications. I would suggest to include in the manuscript the new results regarding chromatin accessibility at longer times following EMT.

I would like to thank Reviewer 1 for his/her excellent suggestions and the time he/she has dedicated to the revision of the manuscript.

While understanding the suggestion of the Reviewer, in our opinion the experiments about chromatin accessibility at longer times are not directly related to the manuscript message. This set of data is related to long-time TGFbeta treatments instead of short-time TGFbeta response, used in the rest of the paper. For example, these experiments are not accompanied by histone modification data at this longer times and therefore complete and appropriated conclusions cannot be deduced. In addition, these results have raised new hypothesis about the reversion of EMT, the state of enhancers during late EMT and its similarity to stem cells, that is not directly related to early enhancers regulation by TGFbeta and should be tested and investigated before being exposed and discussed. Furthermore, we consider that the manuscript contains a large amount of data already, and the inclusion of more data could affect its intelligibility. Finally, the inclusion of these additional data would require to increase the manuscript length considerably, which is slightly over the limit already, and one additional main figure with several panels would have to be included. Therefore, after these considerations we have decided not to include this additional experiments in the final version.

Reviewer #2 (Remarks to the Author):

The authors did an outstanding job addressing all my points. I don't have any additional questions. Great work!

I would like to thank Reviewer 2 for his/her excellent suggestions and the time he/she has dedicated to the revision of the manuscript.

Reviewer #3 (Remarks to the Author):

The authors report on a genome-wide characterization of the genomic response to TGF- β using NMuMG cells, a model cell line for TGF- β -induced epithelial-mesenchymal transition. The genome-wide characterization of the enhancers reveals a substantial response, involving increased chromatin accessibility of a substantial portion of the enhancers throughout the genome, coregulation of genes adjacent to these enhancers, and key roles of AP1 (Jun-Fos) complexes in cooperation with Smad complexes.

Overall: This is an extensive analysis that combines Chip-Seq, RNA-Seq and bioinformatics approaches to characterize the genome-wide response, thus generating a database of genomic changes that occur in response to TGF- β . This information is likely to serve as a resource for further analyses by others. I am very familiar with the biology of TGF- β , including mechanisms of TGF- β signaling and Smad- (and non-Smad-) mediated transcription responses, and reviewed the revised manuscript from that perspective. The authors have satisfactorily addressed my comments, both in the rebuttal and in the revised manuscript.

I would like to thank Reviewer 3 for his/her excellent suggestions and the time he/she has dedicated to the revision of the manuscript.

Some minor comments remain:

- The term TAD known to most in the fields of genomic analyses has been extensively used, but has not been explained (not even what the abbreviation stands for).

TAD abbreviation has been not spelled out both in the Abstract and in the Results section.

- In the Introduction (line 76), the TGF- β receptors are mentioned as serine-threonine kinases; however, they are dual specificity kinases as has been well established starting 26-27 year ago. I am obviously aware that many (some of them stubbornly) do not seem to realize this and propose them to be Ser/Thr kinases, because they phosphorylate R-Smads on Ser.

Thank you very much for this remark. We have now corrected the sentence and added a specific reference.

- line 445” “stablishes” should be “establishes”.

Corrected as indicated by the Reviewer.

- From time to time the authors talk about Smad2/3 occupation of enhancers. They obviously do so because the antibody used recognizes Smad2 and

Smad3. But let's be realistic, most TGF- β responses (with very few exceptions) are Smad3-mediated. I realize that one has to be scientifically correct but maybe it is worth explaining to the reader that Smad3 is the major TGF- β signaling effector, so that the non-aficionado reader does not have misconceptions.

Since the manuscript that originally describes the ChIP-seq data that we have analyzed uses the Smad2/3 terminology and since in our manuscript we cannot discriminate between both factors we have preferred to maintain the Smad2/3 terminology, without further discussion. To explain properly the specific roles of Smad2 and 3 is out of the scope of our manuscript, especially given the text space limitations of the manuscript.

- ref 35 is incomplete.

Corrected as indicated by the Reviewer.